# Inferring multi-scale neural mechanisms with brain network modelling

**Michael Schirner[1,2,3], Anthony Randal McIntosh[4], Viktor Jirsa[5], Gustavo Deco[6,7,8,9], Petra Ritter[1,2,3,10]***

[1]Charité – Universitätsmedizin Berlin, corporate member of Freie Universität Berlin, Humboldt-Universität zu Berlin, and Berlin Institute of Health, Department of Neurology, Berlin, Germany; [2]Berlin Institute of Health (BIH), Berlin, Germany; [3]Bernstein Focus State Dependencies of Learning & Bernstein Center for Computational Neuroscience, Berlin, Germany; [4]Rotman Research Institute of Baycrest Centre, University of Toronto, Toronto, Canada; [5]Institut de Neurosciences des Systèmes UMR INSERM 1106, Aix-Marseille Université Faculté de Médecine, Marseille, France; [6]Center for Brain and Cognition, Computational Neuroscience Group, Department of Information and Communication Technologies, Universitat Pompeu Fabra, Barcelona, Spain; [7]Institució Catalana de la Recerca i Estudis Avançats (ICREA), Barcelona, Spain; [8]Department of Neuropsychology, Max Planck Institute for Human Cognitive and Brain Sciences, Leipzig, Germany; [9]School of Psychological Sciences, Monash University, Melbourne, Australia; [10]Berlin School of Mind and Brain & MindBrainBody Institute, Humboldt University, Berlin, Germany

**Abstract** The neurophysiological processes underlying non-invasive brain activity measurements are incompletely understood. Here, we developed a connectome-based brain network model that integrates individual structural and functional data with neural population dynamics to support multi-scale neurophysiological inference. Simulated populations were linked by structural connectivity and, as a novelty, driven by electroencephalography (EEG) source activity. Simulations not only predicted subjects' individual resting-state functional magnetic resonance imaging (fMRI) time series and spatial network topologies over 20 minutes of activity, but more importantly, they also revealed precise neurophysiological mechanisms that underlie and link six empirical observations from different scales and modalities: (1) resting-state fMRI oscillations, (2) functional connectivity networks, (3) excitation-inhibition balance, (4, 5) inverse relationships between $\alpha$-rhythms, spike-firing and fMRI on short and long time scales, and (6) fMRI power-law scaling. These findings underscore the potential of this new modelling framework for general inference and integration of neurophysiological knowledge to complement empirical studies.
DOI: https://doi.org/10.7554/eLife.28927.001

*For correspondence:
petra.ritter@charite.de

Competing interests: The authors declare that no competing interests exist.

## Introduction

Empirical approaches to characterizing the mechanisms that govern brain dynamics often rely on the simultaneous use of different acquisition modalities. These data can be merged using statistical models, but the inferences are constrained by information contained in the different signals, rendering a mechanistic understanding of neurophysiological processes elusive. Brain simulation is a complementary technique that enables inference on model parameters that reflect mechanisms that underlie emergent behavior, but that are hidden from direct observation (*Breakspear, 2017*).

Brain network models are dynamical systems of coupled neural mass models for simulating large-scale brain activity; coupling is often mediated by estimations of the strengths of anatomical

**eLife digest** Neuroscientists can use various techniques to measure activity within the brain without opening up the skull. One of the most common is electroencephalography, or EEG for short. A net of electrodes is attached to the scalp and reveals the patterns of electrical activity occurring in brain tissue. But while EEG is good at revealing electrical activity across the surface of the scalp, it is less effective at linking the observed activity to specific locations in the brain.

Another widely used technique is functional magnetic resonance imaging, or fMRI. A patient, or healthy volunteer, lies inside a scanner containing a large magnet. The scanner tracks changes in the level of oxygen at different regions of the brain to provide a measure of how the activity of these regions changes over time. In contrast to EEG, fMRI is good at pinpointing the location of brain activity, but it is an indirect measure of brain activity as it depends on blood flow and several other factors. In terms of understanding how the brain works, EEG and fMRI thus provide different pieces of the puzzle. But there is no easy way to fit these pieces together.

Other areas of science have used computer models to merge different sources of data to obtain new insights into complex processes. Schirner et al. now adopt this approach to reveal the workings of the brain that underly signals like EEG and fMRI.

After recording structural MRI data from healthy volunteers, Schirner et al. built a computer model of each person's brain. They then ran simulations with each individual model stimulating it with the person's EEG to predict the fMRI activity of the same individual. Comparing these predictions with real fMRI data collected at the same time as the EEG confirmed that the predictions were accurate. Importantly, the brain models also displayed many features of neural activity that previously could only be measured by implanting electrodes into the brain.

This new approach provides a way of combining experimental data with theories about how the nervous system works. The resulting models can help generate and test ideas about the mechanisms underlying brain activity. Building models of different brains based on data from individual people could also help reveal the biological basis of differences between individuals. This could in turn provide insights into why some individuals are more vulnerable to certain brain diseases and open up new ways to treat these diseases.

DOI: https://doi.org/10.7554/eLife.28927.002

connections based on diffusion-weighted MRI data (so-called structural connectivity or 'connectomes'). Here, we develop a novel type of brain network model, dubbed 'hybrid model', where each subject's EEG data is used to drive neural mass dynamics (*Figure 1*). In brief, we were able to use the resulting hybrid models to reproduce ongoing subject-specific fMRI time series over a period of 20 min and a variety of other empirical phenomena (*Figure 2*). In contrast to previous brain network models that used noise as input, hybrid models are driven by EEG source activity (i.e. EEG sensor activity mapped onto cortical locations) and therefore simultaneously incorporate structural and functional information from individual subjects (*Figure 1*). The injected EEG source activity serves as approximation of excitatory synaptic input currents (EPSCs), which helps to increase the biological plausibility of generated model activity (*Buzsáki et al., 2012*; *Haider et al., 2016*; *Isaacson and Scanziani, 2011*; *Nunez and Srinivasan, 2006*). Individualized hybrid models yield predictions of ongoing empirical subject-specific resting-state fMRI time series (*Figure 3*). Additionally, several empirical phenomena from different modalities and temporal scales are reproduced: spatial topologies of fMRI functional connectivity networks (*Figure 4*), excitation-inhibition (E/I) balance of synaptic input currents, the inverse relationship between $\alpha$-rhythm phase and spike-firing on short time scales (*Figure 5*), and the inverse relationship between $\alpha$-band power oscillations and spike-firing, respectively fMRI oscillations, on long time scales (*Figure 6*), and fMRI power-law scaling (*Figure 7*). More importantly, our subsequent analysis of intrinsic model activity reveals neurophysiological processes that could explain how brain networks produce the aforementioned signal patterns (*Figures 5–7*). That is, simulation results not only predict ongoing subject-specific resting-state fMRI time series and several empirical phenomena observed with invasive electrophysiology methods, but more importantly, they also show how the network interaction of neural populations leads to the

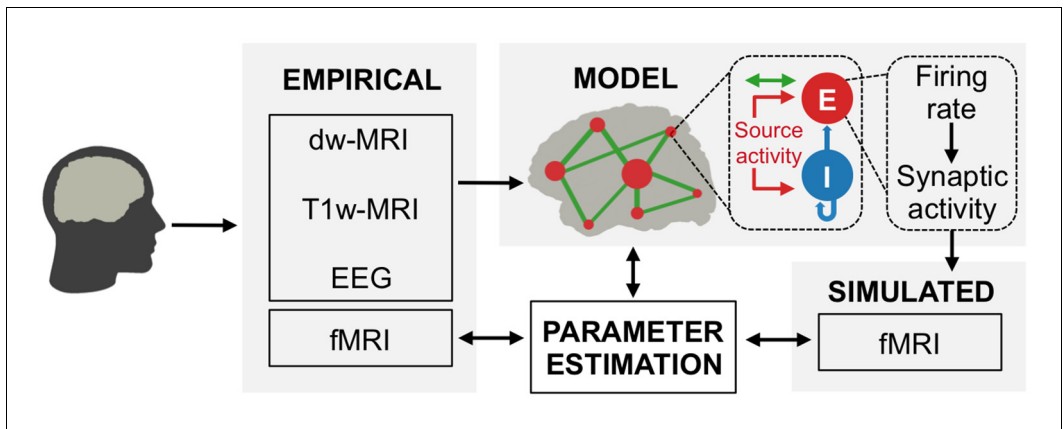

**Figure 1.** Hybrid modeling framework. Hybrid brain network models are constructed from diffusion-weighted MRI tractography and region parcellations obtained from anatomical MRI. The nodes of the hybrid models are injected with subject-specific EEG source activity time series instead of noise. Predicted fMRI time series are fit to each subject's empirical fMRI time series, which were simultaneously acquired with EEG. At each node (small red circles) of the long-range network (green lines) are local networks of excitatory (E) and inhibitory (I) neural population models that are driven by EEG source activity (red arrows). Nodes are globally coupled by structural connectomes (green arrows) that represent the heterogeneous white matter coupling strengths between different brain areas. Synaptic input currents (*Equations 1 and 2*), firing rates (*Equations 3 and 4*) and synaptic activity (*Equations 5 and 6*) underlying fMRI predictions are analysed to identify how neural population activity and network interactions relate to observable neuroimaging signals. See also *Video 1* for a visualization of brain network model construction and exemplary results from hybrid model simulations.

DOI: https://doi.org/10.7554/eLife.28927.003

emergence of these phenomena and how they are connected across multiple temporal scales in a time scale hierarchy.

Resting-state fMRI studies identified so-called 'resting-state networks' (RSNs), which are widespread networks of coherent activity that spontaneously emerge across a variety of species in the absence of an explicit task (*Biswal et al., 1995*; *Fox and Raichle, 2007*; *Raichle et al., 2001*). Despite correlations between fMRI and intracortical recordings (*He et al., 2008*; *Logothetis et al., 2001*), EEG (*Becker et al., 2011*; *Goldman et al., 2002*; *Mantini et al., 2007*; *Moosmann et al., 2003*; *Ritter et al., 2009*) and magnetoencephalography (*Brookes et al., 2011*; *de Pasquale et al., 2010*) the link between RSNs and electrical neural activity is not fully understood. A prominent feature of electrical neural activity are oscillations in the α-band, which is rhythmic activity in the 8 to 12 Hz frequency range first discovered by Hans Berger (*Berger, 1929*). A growing body of research indicates that changes in information processing, attention, perceptual awareness, and cognitive performance are accompanied by rhythmic modulation of α-power and phase (*Busch et al., 2009*; *Klimesch, 1999*; *Mathewson et al., 2009*). The observed inverse relationship between α-band activity and neural firing is central to hypotheses on its functional significance termed 'gating by inhibition' and 'pulsed inhibition' (*Jensen and Mazaheri, 2010*; *Klimesch et al., 2007*). Interestingly, intracellular recordings showed that inhibitory events are inseparable from excitatory events, resulting in an ongoing excitation-inhibition balance (E/I balance) (*Isaacson and Scanziani, 2011*; *Okun and Lampl, 2008*). The significance of the α-rhythm is underscored by strong negative correlations between ongoing α-band power fluctuation and resting-state fMRI amplitude fluctuation (*de Munck et al., 2008*; *Feige et al., 2005*; *Goldman et al., 2002*; *Moosmann et al., 2003*). Lastly, despite wide-spread interest in critical dynamics (*Bak, 2013*), the key determinants of emergent power-law scaling, a signal pattern that is ubiquitous in nature and commonly observed in neural activity, are unclear (*Beggs and Timme, 2012*; *Marković and Gros, 2014*).

To illustrate the potential of this framework for inference of neurophysiological processes, we show inferred mechanisms for three different empirical phenomena and how they relate to other well-established neural signal patterns (*Figure 2*). Upon finding that the hybrid model predicts fMRI activity, we first sought to identify how injected EEG drove the prediction of subject-specific fMRI

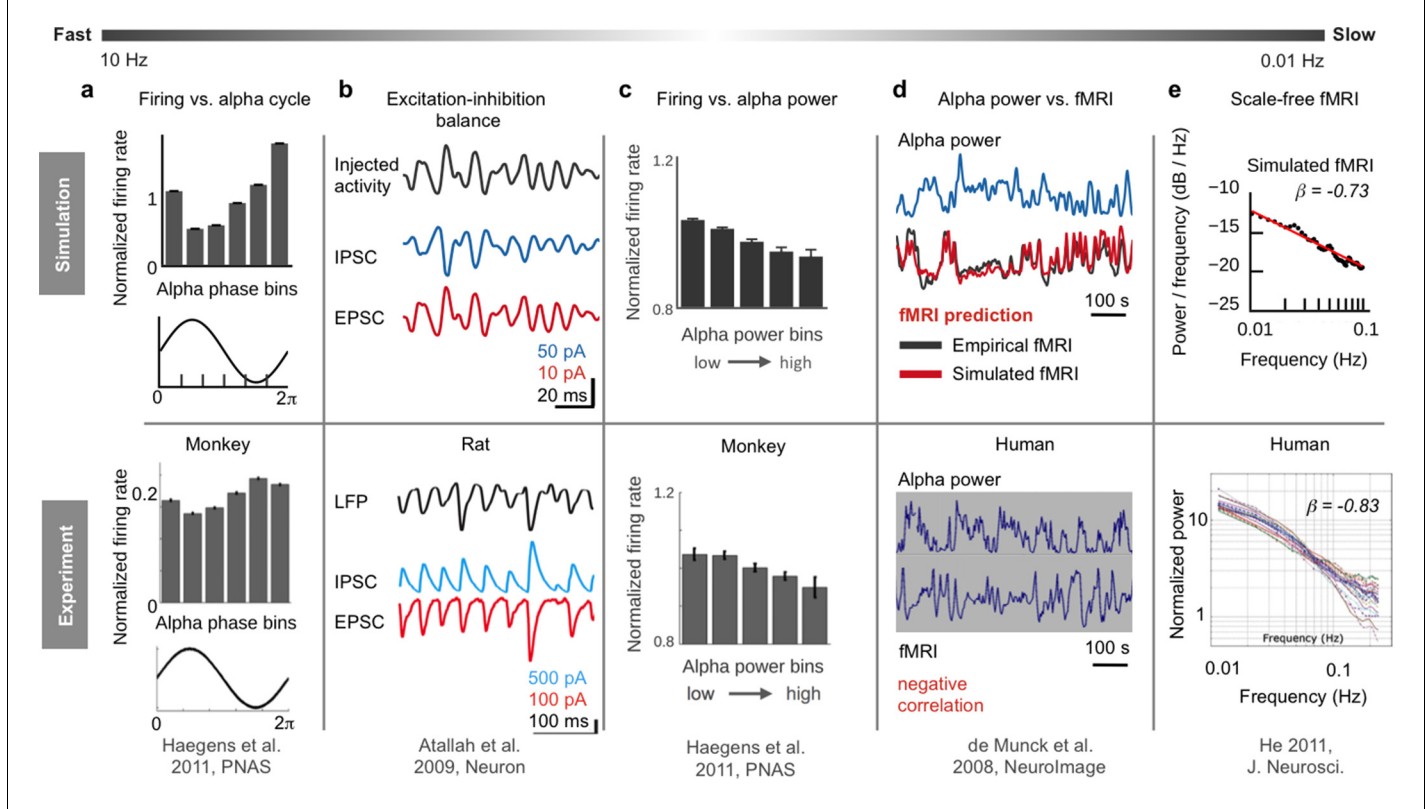

**Figure 2.** Overview of six empirical phenomena on different temporal scales reproduced by the hybrid model. (**a**) Neuron firing is inversely related to the phase of α-waves: during peaks of α-waves, neurons fire the least, while they fire maximally during troughs (adapted from *Haegens et al., 2011*). (**b**) Our simulations indicate that the inverse relationship between firing and α-phase is related to the ongoing balancing of neural excitation and inhibition (adapted from *Atallah and Scanziani, 2009*). Reprinted with permission of Elsevier.). (**c**) On a longer time scale (<0.25 Hz), oscillations of firing rates are inversely related to α-band power fluctuations (adapted from *Haegens et al., 2011*). (**d**) Model simulations suggest a mechanism that transforms α-band power oscillations into fMRI oscillations, predicting subject-specific resting-state fMRI time series, corresponding spatial network patterns and the inverse correlation between α-power and fMRI (adapted from *de Munck et al., 2008*. Reprinted with permission of Elsevier.). (**e**) Emergence of scale-free fMRI power spectra (adapted from *He, 2011*) resulted from long-range network input. (**f**) Individual functional connectivity matrices were predicted over long and short time windows (adapted from *Allen et al., 2014*).
DOI: https://doi.org/10.7554/eLife.28927.004

time series. Analysis led us to a mechanism that transformed α-power fluctuations of injected EEG source activity into fMRI oscillations. The identified mechanism may explain the empirically observed correlation between EEG and fMRI on the longer time scale of slow fMRI oscillations. Consequently, we asked how the inhibitory effect of increased α-band power was created on the faster time scale of α-phase fluctuations. Analysis led us to the identification of an inhibitory effect resulting from the interaction of postsynaptic current oscillations and local population circuitry. Interestingly, parameter space exploration showed that prediction quality decreased when long-range coupling was deactivated (i.e. when the nodes of the long-range network were isolated from each other). Therefore, we interrogated the model for the influence of structural coupling on the emergence of fMRI oscillations and found that global coupling amplified brain oscillations in a frequency-dependent manner, amplifying slower oscillations more than faster oscillations, which facilitated the emergence of power-law scaling. Starting with fast time scale effects, our first model outcome accounts for the invasively observed inverse relationship between spike-firing and α-rhythm phase by identifying a mechanism that relates this phenomenon to ongoing E/I balance. The second model outcome posits a neural origin of fMRI RSN oscillations by identifying an explicit mechanism that transforms ongoing α-power fluctuations into slow fMRI oscillations, which also explains the empirically observed anti-correlations between α-power and fMRI time series. Our third model outcome indicates that scale

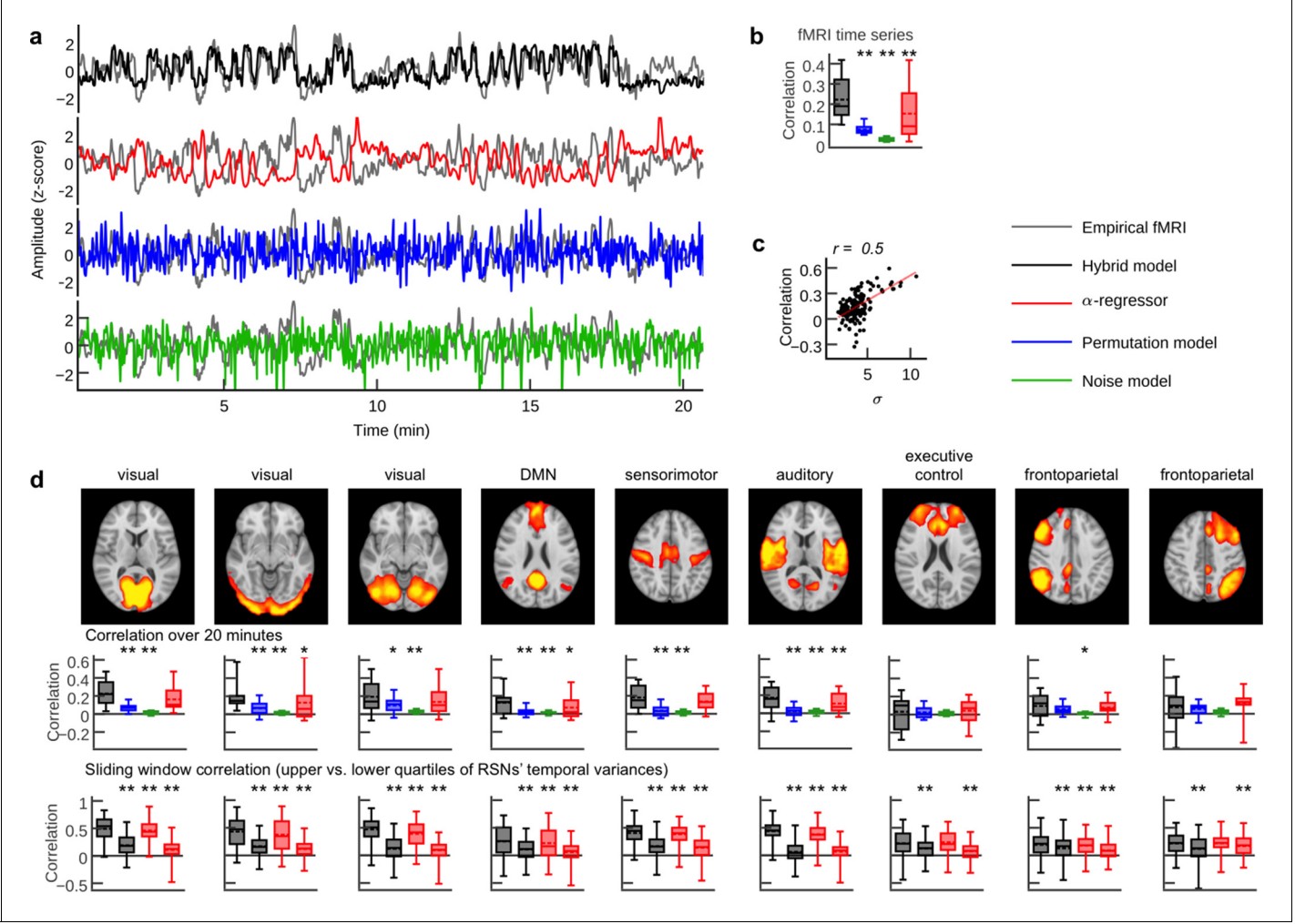

**Figure 3.** Person-specific fMRI time series prediction. (a) Example time series of the hybrid model and the three control scenarios from one subject. (b) Box plots of average correlation coefficients between all simulated and empirical region time series (20.7 min) for each subject (n = 15; α-regressor values were inverted for illustration purposes). (c) Scatter plot of RSN time course standard deviation (s.d.) versus prediction quality. Dots depict data from the nine RSN time courses for each subject. (d) Comparison of prediction quality during upper versus lower quartile of epoch-wise RSN time course s.d.s. Upper row: spatial activation patterns of nine RSNs. Middle row: correlation coefficients between RSN temporal modes and hybrid model simulation results and the three control scenarios. Lower row: sliding window (length: 100 fMRI scans = 194 s; step width: one fMRI scan) correlations for the upper (first and third boxplot per panel) and lower quartiles (second and fourth boxplot per panel) of window-wise RSN temporal mode for the hybrid model and the α-regressor. Asterisks indicate significantly increased prediction quality of the hybrid model compared to control scenarios in one-tailed Wilcoxon rank sum test (*p<0.05, **p<0.01). Additionally, all hybrid model correlations in (b) and (d) were tested for the null hypothesis that they come from a distribution whose median is zero at the 5% significance level. All tests rejected the null hypothesis of zero medians except for RSN correlations over 20 min for the executive control and the frontoparietal networks (middle row).

DOI: https://doi.org/10.7554/eLife.28927.005

The following figure supplement is available for figure 3:

**Figure supplement 1.** Parameter space exploration results.

DOI: https://doi.org/10.7554/eLife.28927.006

invariance of fMRI power spectra results from self-reinforcing feedback excitation via long-range structural connectivity, which leads to frequency-dependent amplification of neural oscillations.

In summary, our biophysically grounded brain model has the potential to test mechanistic hypotheses about emergent phenomena such as scale-free dynamics, the crucial role of excitation-inhibition balance and the haemodynamic correlates of α-activity. However, there is another perspective on this form of hybrid modeling. Because it uses empirical EEG data to generate predictions of fMRI responses, it can be regarded as a form of multimodal fusion under a generative model that is both

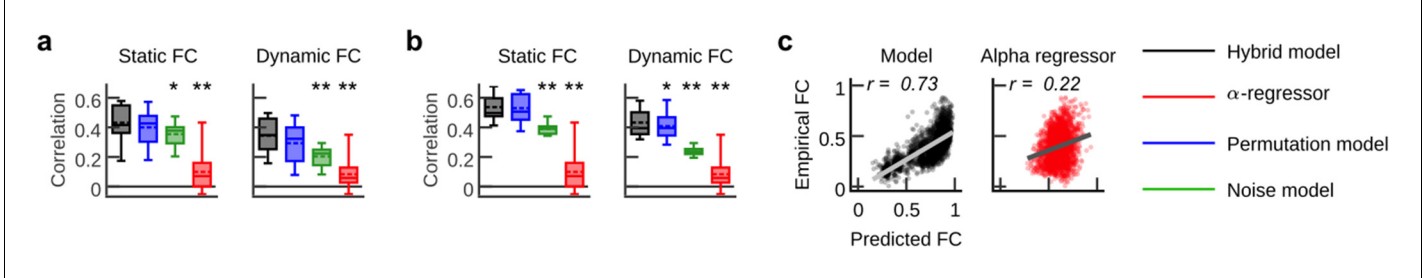

**Figure 4.** Functional connectivity prediction. (a, b) Box plots show correlation coefficients obtained from correlating all subdiagonal entries of empirical and simulated FC matrices. FC was computed for long epochs (static FC; computed over 20.7 min) and short epochs (dynamic FC; average sliding window correlation; 100 fMRI scans window length; one fMRI scan step width). Results were compared for (a) the parameter set that generated the best fMRI time series prediction and (b) the parameter set that yielded the best FC predictions for each subject. (c) Scatter plots compare empirical and simulated average FC for hybrid model simulations and the α-regressor. Dots depict all pair-wise region time series correlations averaged over all subjects. Asterisks in (a) and (b) indicate significantly increased prediction quality of the hybrid model compared to control conditions in one-tailed Wilcoxon rank sum test (*p<0.05, **p<0.01).

DOI: https://doi.org/10.7554/eLife.28927.007

physiologically and anatomically grounded. In addition, because we use connectivity constraints based on tractography, it also serves to fuse structural with functional data.

## Results

### Hybrid models predict subject-specific fMRI time series

The used brain network models are dynamical systems where individual brain areas are simulated by coupled neural mass models. Long-range coupling was weighted by heterogeneous strength estimates obtained from white-matter tractography, a method that estimates neural tracts from diffusion-weighted MRI data. The used neural mass models approximate the average ensemble behaviour of networks of spiking neuron models and were derived in a previous study (*Deco et al., 2013*) using a dynamic mean-field technique (*Deco et al., 2008*; *Wong and Wang, 2006*). In

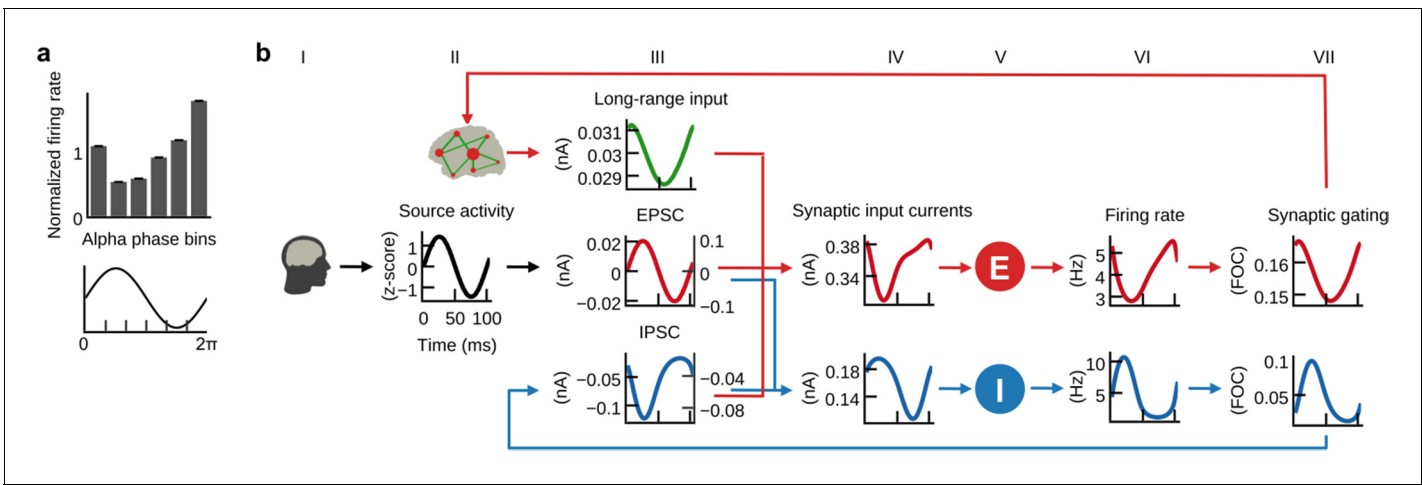

**Figure 5.** E/I balance generates the inverse relationship between α-phase and firing. (a) Histogram of population firing rates divided into six bins according to α-cycle segments and normalized relative to the mean firing rate of each cycle. Population firing rates were highest during the trough and lowest during the peak of α-cycles. (b) Grand average waveforms of population inputs and outputs time locked to α-cycles of injected EEG source activity (black, column II). Left and right axes denote input currents to excitatory and inhibitory populations, respectively. Please refer to the main text for a description of the mechanism that explains the inverse relationship between α-cycles and firing rates.

DOI: https://doi.org/10.7554/eLife.28927.008

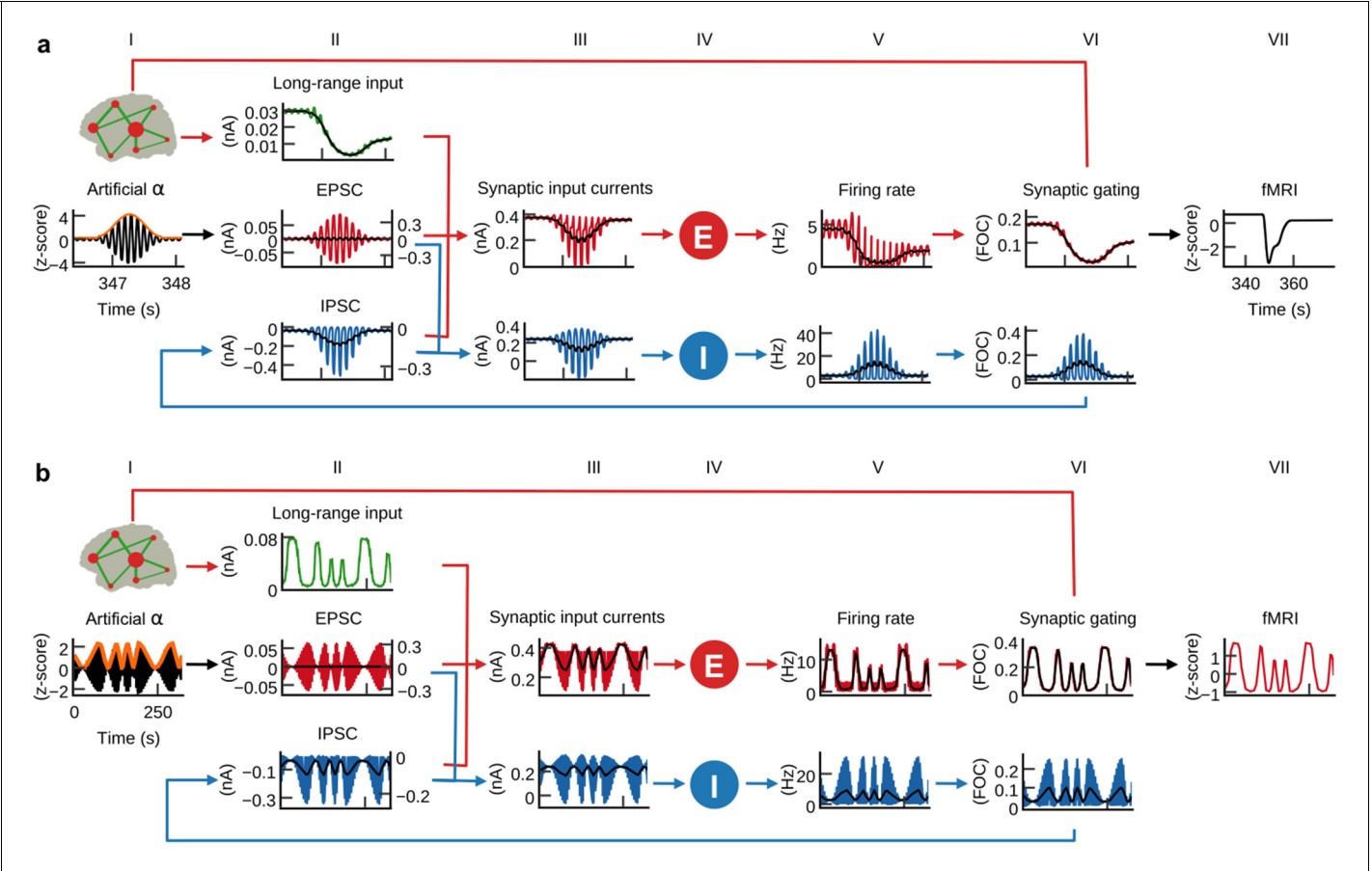

**Figure 6.** α-power fluctuations generate fMRI oscillations. Grand average waveforms of population inputs and outputs on longer time scales. (**a**) Hybrid models were injected with artificial α-activity consisting of 10 Hz sine oscillations that contained a single brief high-power burst (black, column I; orange: signal envelope). While positive deflections of the α-wave generated positive deflections of inhibitory population firing rates, large negative deflections were bounded by the physiological constraint of 0 Hz (blue, fifth column; black: moving average). (**b**) Hybrid models were injected models with 10 Hz sine waves where ongoing power was modulated similar to empirical α-rhythms (0.01–0.03 Hz). Similarly to (**a**), but for a longer time frame, inhibitory populations rectified negative deflections, which introduced the α-power modulation as a new frequency component into firing rates and fMRI time series.

DOI: https://doi.org/10.7554/eLife.28927.009

The following figure supplement is available for figure 6:

**Figure supplement 1.** α-power predicts firing rate.

DOI: https://doi.org/10.7554/eLife.28927.010

contrast to previous brain network models that used noise as input, the neural mass models of our 'hybrid' model are driven by EEG source activity that was simultaneously acquired with fMRI (*Figure 1*). Simulation results predicted a considerable part of the variance of ongoing subject-specific resting-state fMRI time series (*Figure 3*) and spatial network topologies, that is, fMRI functional connectivity (*Figure 4*; functional connectivity is here defined as the pair-wise correlation matrix between region time series). Furthermore, fitted models reproduced a variety of empirical phenomena observed with EEG and invasive electrophysiology (*Figure 2*) and, more importantly, simulation results revealed mechanistic explanations for the emergence of these phenomena (*Figures 5*, *6* and *7*).

We constructed individual hybrid brain network models for 15 human adult subjects using each subject's own structural connectomes and injected each with their own region-wise EEG source activity time courses that were acquired simultaneously with the fMRI data subsequently predicted. Using exhaustive searches, we tuned three global parameters for each of the 15 individual hybrid brain network models to produce the highest fit between each of the subject's empirical region-

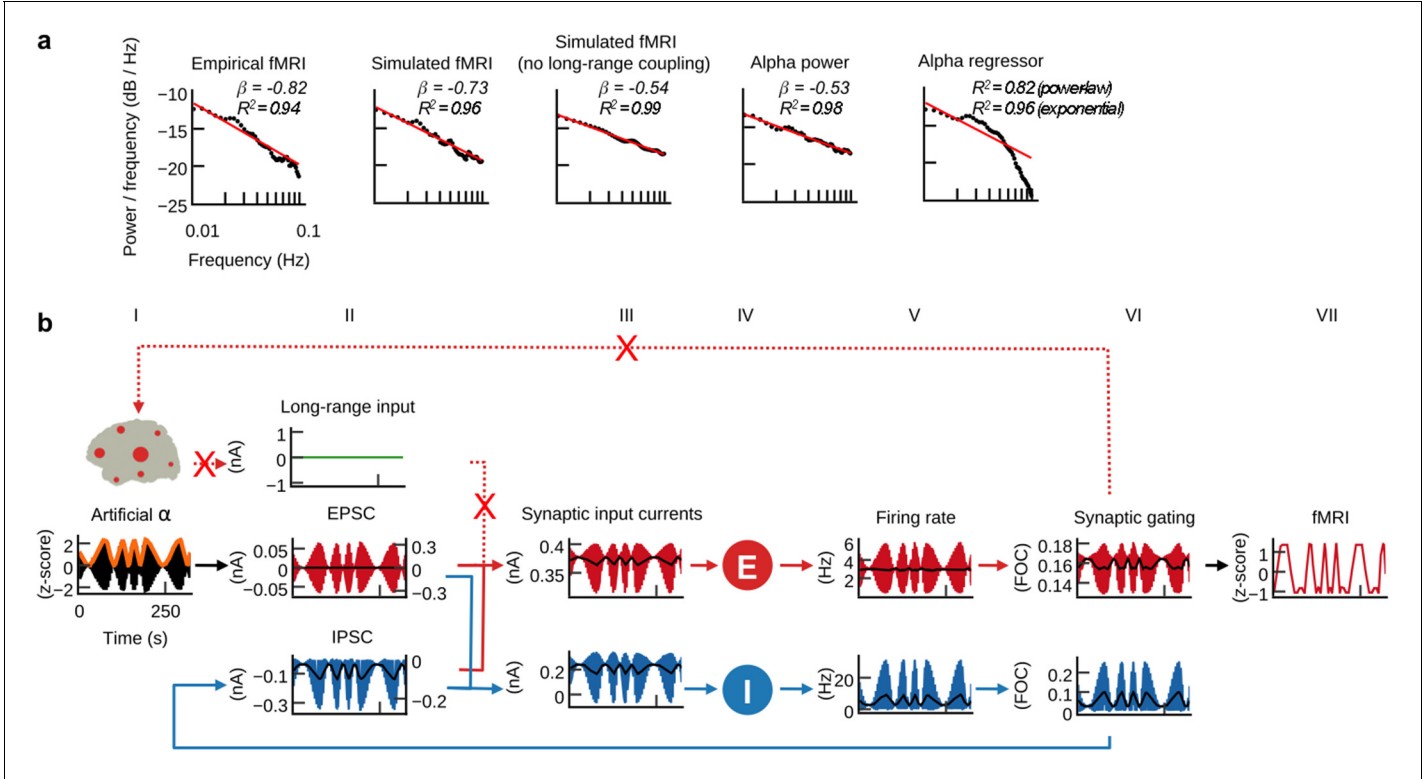

**Figure 7.** Long-range coupling controls fMRI power-law scaling. (a) Power spectral densities of simulated and empirical fMRI and empirical α-band activity (straight-line fits of power spectra are for illustration purposes only; scale-invariance was determined in the time domain using rigorous model selection criteria, see Materials and methods). (b) As in *Figure 6b*, but with disabled long-range coupling. In contrast to *Figure 6b*, the amplitudes of firing rates, synaptic gating and fMRI are equally large, while in *Figure 6b* amplitudes were larger during slower α-band power modulations.

DOI: https://doi.org/10.7554/eLife.28927.011

The following figure supplements are available for figure 7:

**Figure supplement 1.** As in *Figure 7a*, but simulations were performed with the simplified hybrid model, that is, the models used no feedback inhibition control (FIC) and a single value for all local inhibitory connections $J_i$ was used.

DOI: https://doi.org/10.7554/eLife.28927.012

**Figure supplement 2.** Fine-grained parameter space exploration of the simplified hybrid model for an exemplary subject.

DOI: https://doi.org/10.7554/eLife.28927.013

average fMRI time series and corresponding simulated time series (*Figure 3—figure supplement 1*). Our motivation for choosing that parameter set that produced the highest correlation between simulated and empirical fMRI time series is based on our goal to infer the underlying (but unobservable) dynamics and parameters of the real system. This idea is based on the assumption that when the model optimally fits observable brain activity, then also the underlying unobservable brain activity is faithfully reproduced. The first parameter scales the global strength of long-range coupling between regions. The second and third parameters scale the strengths of EEG source activity inputs injected into excitatory and inhibitory populations, respectively. To better assess the quality of fMRI predictions, we compared hybrid model results with three control scenarios: (i) the original noise-driven brain network model, (ii) a variant of the hybrid model where the time steps of the injected EEG source activity time series were randomly permuted and (iii) a statistical model where the ongoing α-band power fluctuation of injected EEG source activity was convoluted with the canonical hemodynamic response function (henceforth called α-regressor). The first two controls are brain network models and the third is inspired by traditional analyses of empirical EEG-fMRI data. The controls serve to exclude that the obtained correlations between simulated and empirical fMRI is a trivial outcome that would also be produced by the original noise-driven model or with random input time series.

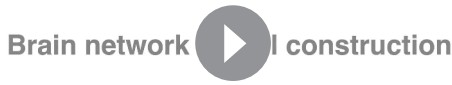

**Video 1.** Reverse-engineering neural information processing. The video shows how computational brain network models are constructed from individual neuroimaging data, how these models can be used to simulate different types of neural activity of individual subjects on multiple temporal scales and how model activity can be used to derive mechanisms of brain function.

DOI: https://doi.org/10.7554/eLife.28927.014

Visual inspection of example time series showed good reproduction of characteristic slow (<0.1 Hz) RSN oscillations by the hybrid model and the α-regressor (albeit inverted for the latter), but poor reproduction of temporal dynamics in the case of noise and random permutations models (*Figure 3*). We compared the average correlation coefficients between all simulated and empirical fMRI time series between all four scenarios (i.e. hybrid model and the three control setups). Predictions from the hybrid model correlated significantly better with empirical fMRI time series than predictions from the two random models and the α-regressor (*Figure 3b*). For the hybrid model, five-fold cross-validation showed no significant difference of prediction quality between training and validation data sets (two-tailed Wilcoxon rank sum test, p=0.71, $t = 0.54$, Cliff's delta $d = 0.0044$) and between validation data sets and prediction quality for the full time series (two-tailed Wilcoxon rank sum test, p=0.42, $t = -0.2$, Cliff's delta $d = -0.067$).

To estimate the ability of the four scenarios to predict the time courses of different commonly observed RSNs we performed a group-level spatial independent component analysis (ICA) of the empirical fMRI data. Next, we computed average correlation coefficients between each subject-specific RSN time course and the model regions at the position of the respective RSN. As in the case of region-wise fMRI (*Figure 3b*), correlation coefficients of the hybrid model were significantly larger than the control network models for most RSNs (*Figure 3d*). The sliding-window analyses showed that prediction quality varied over time, regions and subjects: window-wise prediction quality was highly correlated with the standard deviation of RSN temporal modes (*Figure 3c,d*). That is, the higher the variance contributed to overall fMRI activity by an RSN in a given subject and time window, the better the prediction of empirical fMRI, which might reflect increased synchrony of electrical activity (see Discussion). As a consequence, epochs in the upper quartile of RSN s.d.s were significantly better predicted than epochs in the lower quartile (*Figure 3d*). In order to assess the subject-specificity of fMRI time series predictions, we correlated all simulation results (i.e. for every subject and every tested parameter combination) also with the empirical fMRI activity of all other subjects. We found that the maximum correlation coefficients over all tested parameters were significantly larger when empirical and simulated data sets belonged to the same subject compared to when they came from different subjects (p<<0.01, Wilcoxon rank sum test).

Next, we compared the ability of all four setups to predict the spatial topology of empirical fMRI networks. In contrast to time series prediction, the α-regressor showed low correlations with empirical functional connectivity (FC). Compared to the α-regressor, all three model-based approaches provided significantly better predictions of subjects' individual long-epoch FC and short-epoch FC (*Figure 4*). Furthermore, hybrid model simulation results correlated significantly better with empirical network topology than predictions obtained from the noise-driven model (*Figure 4a,b*). Interestingly, correlations for hybrid and random permutation models were effectively the same, likely because the long-range network dynamics, which drive the emergence of FC by structural coupling, would be relatively preserved when permuting injected activity. Prediction of group-average FC (all pairwise FC values averaged over all subjects) was better for the hybrid model compared to the α-regressor (*Figure 4c*).

## E/I balance generates the inverse relationship between α-phase and firing

After fitting the individual hybrid models for each of the 15 subjects, we analyzed the local population activity to infer neurodynamic mechanisms underlying predicted fMRI time series. We found that on the fast time scale of individual α-cycles (~100 ms) the optimized hybrid model reproduced the inverse relationship between α-phase and firing rates observed in invasive recordings (*Haegens et al., 2011*) (*Figure 5a*). To investigate these fast-acting dynamics related to α-phase, we computed grand average waveforms of modeled synaptic inputs, population firing rates, and synaptic gating time-locked to the zero-crossings of α-cycles. Resulting waveforms illustrate how the ongoing balancing of excitatory and inhibitory inputs generated the inverse relation between α-oscillations and neural firing (*Figure 5b*).

In hybrid models, individual subject's (*Figure 5*, column I) source activity (column II) is used as an approximation of EPSCs (column III). As a result of optimizing the three model parameters, EPSCs dominated the sum of synaptic input current to inhibitory populations (column IV). Consequently, inhibitory populations' (column V) firing rates (column VI) and synaptic gating (column VII) closely followed the shape of EPSCs. Because of the monotonic relationship between input currents and output firing rates (defined by *Equation 3 and 4*), the waveform of inhibitory firing rates and synaptic gating also closely followed injected EPSCs. As increased input to inhibitory populations leads to increased inhibitory effect and vice versa, resulting feedback inhibition waveforms (IPSC, column III) were inverted to EPSCs. Furthermore, the amplitude fluctuation of EPSCs and IPSCs was proportional. That is, stronger EPSCs preceded and helped to generate stronger IPSCs. In other words, excitation and inhibition were balanced during each cycle, which is in accordance with published electrophysiology results (*Haider et al., 2016*; *Isaacson and Scanziani, 2011*; *Okun and Lampl, 2008*; *Xue et al., 2014*). Consequently, IPSCs peaked during the trough of the α-phase and were lowest during the peak of the α-phase. Fitting the models to fMRI activity resulted in a biologically plausible ratio of EPSCs to IPSCs (*Xue et al., 2014*), with IPSC amplitudes being about three times larger than EPSC amplitudes (compare left axes of EPSC and IPSC plots). Because IPSCs have dominated excitatory population inputs, excitatory populations' firing rates showed a similar shape as IPSCs, that is, they peaked during the trough of the α-cycle and fell to their minimum during the peak of the α-cycle, thereby reproducing the empirical relationship between α-cycle and firing rate (*Haegens et al., 2011*). Columns IV, VI and VII refer to *Equations 1, 3 and 5* (excitatory population) and 2, 4 and 6 (inhibitory population), respectively.

In summary, the fast population activity underlying fMRI predictions showed a rhythmic modulation of firing rates on the fast time scale of individual α-cycles in accordance with empirical observations (*Haegens et al., 2011*). Analyses revealed that periodically alternating states of excitation and inhibition resulted from the ongoing balancing of EPSCs by feedback IPSCs, which explains α-phase-related neural firing.

## α-power fluctuations generate fMRI oscillations

Similar to intracranial recordings in monkey (*Haegens et al., 2011*), we found that increased α-power of injected EEG source activity was accompanied by decreased firing rates (*Figure 6—figure supplement 1*). Furthermore, we also observed the empirically observed inverse relationship between α-power and fMRI amplitude (*Goldman et al., 2002*; *Moosmann et al., 2003*) in our empirical data in the form of negative correlations between the α-regressor and fMRI activity (*Figure 3*). Our findings raised the question what physiological mechanism led to this inverse relationship between α-power and firing rate, respectively, fMRI amplitude. We therefore analyzed model activity on the longer time scale of α-power fluctuations. To isolate the effects of α-waves from other EEG rhythms, we replaced the injected EEG-source activity in the 15 individual hybrid models with artificial α-activity (*Figure 6a*, column I) and simulated all 15 hybrid models using the single parameter set that previously generated the highest average fMRI time series prediction quality (*Figure 3—figure supplement 1*). Injected activity consisted of a 10 Hz sine wave that contained a single brief high-power burst in its center in order to allow for model activity to stabilize for sufficiently long phases before and after the high-power burst. After simulation, we computed grand average waveforms of model state variables over all simulated region time series and found that input currents, firing rates, synaptic activity and fMRI activity of excitatory populations decreased in response to the

α-burst (*Figure 6a*). Notably, this behavior emerged despite the fact that injected activity (column I) was centered at zero, that is, positive and negative deflections of input currents were balanced. The reason for the observed asymmetric response to increasing input α-power levels originated from inhibitory population dynamics: while positive deflections of α-cycles generated large peaks in ongoing firing rates of inhibitory populations, negative deflections were bounded by 0 Hz (column V). Because of this rectification of high-amplitude negative half-cycles, average per-cycle firing rates of inhibitory populations increased with increasing α-power. As a result, also feedback inhibition (IPSC, column II) had increased for increasing α-power, which in turn led to increased inhibition of excitatory populations, decreased average firing rates, synaptic gating variables (column VI) and ultimately fMRI amplitudes (column VII).

We next analyzed the relationship between α-power fluctuations and fMRI oscillations. We generated artificial α-activity consisting of a 10 Hz sine wave that was amplitude modulated by slow oscillations (cycle frequencies between 0.01 and 0.03 Hz) and injected it into the hybrid models of all subjects (*Figure 6b*, column I). As in the previous example, inhibitory populations filtered negative α-deflections during epochs of increased power (column V). This half-wave rectification led to a modulation of average per-cycle firing rates in proportion to α-power. Consequently, the power modulation of the injected α-oscillation was introduced as a new slow frequency component into the resulting time series. The activity of inhibitory populations can be compared to envelope detection used in radio communication for AM signal demodulation. The new frequency component introduced by half-wave rectification of α-activity modulated feedback inhibition (IPSC, column II), which in turn modulated excitatory population firing rates (column V). Furthermore, the resulting oscillation of firing rates was propagated to synaptic dynamics (column VI) where the large time constant of NMDAergic synaptic gating ($\tau_{NMDA}$NMDA100 ms vs. $\tau_{GABA}$GABA10 ms) led to an attenuation of higher frequencies. The low-pass filtering property of the hemodynamic response additionally attenuated higher frequencies such that in fMRI signals (column VII) only the slow frequency components remained, based on the assumption that neurovascular coupling was mediated exclusively by excitatory synaptic activity. To restate: α-power fluctuation introduced an inverted slow modulation of firing rates and synaptic activity; the low-pass filtering properties of synaptic gating and hemodynamic responses attenuated higher frequencies such that only the slow oscillation remained in fMRI signals. To check whether this mechanism is robust to the choice of the frequency of the injected α-rhythm (10 Hz) we simulated otherwise identical models for artificial α-waves at 9 Hz and 11 Hz frequencies and found qualitatively identical results: simulated fMRI and moving average firing rate time series of the 9 Hz and the 11 Hz model had correlation coefficients $r > 0.99$ with the respective time series of the 10 Hz model.

In summary, we found that increased α-power led to increased feedback inhibition of excitatory populations introducing a slow modulation of population firing, which can explain the empirically observed anticorrelation between α-power and fMRI.

## Long-range coupling controls fMRI power-law scaling

Empirical fMRI power spectra follow a power-law distribution $P \propto f^{\beta}$, where $P$ is power, $f$ is frequency and $\beta$ the power-law exponent. In accordance with systematic analyses of empirical data (*He, 2011*), average power spectra of our empirical fMRI data obeyed power-law distributions with exponent $\beta_{emp} = -0.82$ (*Figure 7a* and *Figure 7—figure supplement 1*). We tested for the existence of power-law scaling in the time domain by using rigorous model selection criteria that overcome the limitations of simple straight-line fits to power spectra (see Materials and methods; for illustration purposes straight-line fits are shown in *Figure 7a* and *Figure 7—figure supplement 1*).

Our previous results associated resting-state fMRI oscillations with electrical neural activity by identifying a neural mechanism that transforms α-band power fluctuations into fMRI oscillations (*Figure 6*). This mechanism suggests that EEG α-band power fluctuations are transformed into fMRI amplitude fluctuations. Therefore, it is surprising that the power spectra of wide-band and α-band EEG have considerably smaller negative exponents than empirical fMRI ($\beta_{\alpha-band} = -0.53$ for α-power and $\beta_{wide-band} = -0.47$ for wide-band power). However, in agreement with empirical fMRI, our simulated fMRI had a larger negative exponent ($\beta_{sim} = -0.73$) than the α-band power of the injected EEG source activity ($\beta_{\alpha-band} = -0.53$). This result implies that the power-law slope increased during the process that transformed electrical band-power fluctuations into fMRI amplitude fluctuations. Indeed, comparison of power spectra indicated that simulated fMRI had a higher negative exponent

than EEG source-activity, because the power of slower oscillations increased relative to the power of faster oscillations (*Figure 7a* and *Figure 7—figure supplement 1*). That is, model dynamics transformed synaptic input activity such that the amplitude of output oscillations increased inversely proportional to their frequency. Interestingly, when long-range coupling was deactivated in simulations that used EEG source activity as input, the power-law exponent of simulated fMRI ($\beta_{sim\_Gzero}$ = −0.54) was close to the exponent of the α-band power time course of the injected EEG source activity ($\beta_{\alpha-band}$ = −0.53). The effect was also visible when comparing our previous model simulations that used artificial α-activity (*Figure 6b*), with simulations where long-range coupling was deactivated (*Figure 7b*). When long-range coupling was deactivated, the amplitudes of fMRI oscillations were equally large for all oscillation frequencies (*Figure 7b*, column VII). In contrast, when long-range coupling was activated, with everything else being identical, the amplitudes of slower fMRI oscillations were larger than the amplitudes of faster oscillations (*Figure 6b*, column VII), although the amplitudes of the injected artificial α-band power oscillations were equally large for all oscillations (*Figures 6b* and *7b*, column I). When long-range coupling was present, the amplitudes of slow oscillations increased and the relationship between power and frequency of oscillations approximated the power-law exponent found in empirical fMRI power spectra (*Figure 7a*). With everything else being identical, we concluded that long-range coupling was responsible for increasing the power of slower oscillations relative to faster oscillation.

Comparison of the individual components of population inputs for activated (*Figure 6b*, column II) vs. deactivated (*Figure 7b*, column II) long-range coupling reinforced that the only difference in population inputs between both setups was the shape of long-range input. The amplitudes of long-range input oscillations (*Figure 6b*, column II, green trace) were inversely proportional to the band-power oscillation of injected artificial α-activity. In accordance with the effect of α-band power on population activity that we described earlier, long-range input increased when α-band power decreased, while during epochs of increased α-activity long-range coupling decreased. Consequently, this fluctuation of long-range input was coherent with the fluctuation of IPSCs that resulted from the fluctuation of α-band power, which further amplified the effect of α-band power on population activity. During epochs of low α-activity long-range coupling conveyed feedforward excitation that further reinforced the increasing of firing and synaptic gating. Because of this consensual modulation of input currents, total input currents were increased when α-band power was decreased, which resulted in larger amplitudes of firing rates, synaptic activity and fMRI.

Due to the large time constant of excitatory synaptic gating ($\tau_{NMDA}$ = 100 ms), long-range excitation decayed relatively slowly, which enabled excitatory activity to accumulate and perpetually reinforce within the long-range network. The period of time for which this feedforward excitation persisted was longer during slower oscillations than during faster oscillations. Consequently, synaptic activity (column VI) had more time to accumulate and was therefore larger during slower oscillations compared to faster oscillations. As a result, the amplitudes of excitatory population output (columns V, VI, and VII) reached higher values during slower oscillations than during faster oscillations when long-range coupling was activated (*Figure 6b*). Accordingly, the power of slower oscillations, and therefore the slope of the power spectrum, increases in the case of long-range coupling. Note that this effect (i.e. that slower oscillations reach higher amplitudes) can already be observed in firing rates and synaptic gating time series, which excludes an influence of the hemodynamic forward model. In contrast, in the case of deactivated long-range coupling (*Figure 7b*) all amplitude peaks are approximately equal, which was the expected result, since the amplitude-peaks of the power modulation of injected α-activity were equally high by construction (column I, orange trace).

We asked how the relative strengths of white-matter excitation and feedback inhibition influence power-law scaling. In order to test how E/I balance affects power-law scaling, we varied the strength of long-range coupling and, also globally, the strength of feedback inhibition. That is, in contrast to our previous simulations, the strength of feedback inhibition was controlled by a single parameter for all inhibitory populations. The other parameters, that is the strengths of EEG source activity injected into excitatory and inhibitory populations, were kept fixated. Screening of individual parameter spaces showed that the power-law exponent of simulated fMRI depended on the balance of long-range excitation and local inhibition: the 2D distribution of the prediction quality of fMRI time series, functional connectivity and the power-law exponent showed a characteristic diagonal pattern. That is, increased long-range coupling required increased local feedback inhibition for producing best predictions of fMRI, FC and power-law exponents, which demonstrated the crucial role of E/I

balance for the emergence of scale invariance and long-range correlations (*Figure 7—figure supplement 1* and *Figure 7—figure supplement 2*).

## Discussion

In this work, we describe a biophysically based brain network model that predicts a considerable part of subject-specific fMRI resting-state time series on the basis of concurrently measured EEG. Importantly, we show how this novel modelling approach can be used to infer the neurophysiological mechanisms underlying neuroimaging signals. Instead of mere reproduction of empirical observations, our central aim was to provide an integrative framework that unifies empirical data with theory of the nervous system in order to derive mechanisms of brain function underlying empirical observations across many scales. Clearly, the sequence of analyses and implicit hypothesis testing presented in this paper represents one of many lines of enquiry. The more general point made by this report is that our hybrid model can be used to both test hypotheses and to build hypotheses. In other words, many of the questions (for which we offer answers) only emerged during application of the model, which allowed us to pursue a particular narrative in understanding the genesis of different empirical phenomena. A key point of consideration is that the brain model was built from networks of generic neural population models that were constrained by empirical data, but not explicitly constructed to address specific reproduced phenomena. This is mirrored by the emergence of processes at considerably faster time scales than the subject-specific fMRI time series that were the target of the model fitting. It is important to point out that the inferred mechanisms constitute candidate hypotheses that require empirical falsification. The model-derived mechanisms make concrete predictions on the waveforms of different input currents, output firing rates, synaptic activities and fMRI signals, which can be empirically tested. Through ongoing integration of biological knowledge, falsification with empirical data and subsequent refinement, hybrid brain network models are intended to represent a comprehensive and increasingly accurate theory about large-scale brain function. The construction of hybrid brain network models and our major results are visualized in *Video 1*.

Hybrid models draw on empirically estimated EEG source activity to constrain synaptic input current dynamics. This approach is motivated by the need for a model that not only reproduces static features of brain activity, like functional connectivity, but that produces these features on the basis of biologically plausible time series dynamics. Underlying the approach is the consideration that commonly used fitting targets of BNMs, like FC or power spectral features (e.g. slow BOLD oscillations, EEG $\alpha$-peak), can in principle be generated by time series that are, except from the fitted features, not necessarily biologically plausible. For example, a wide range of waveforms can produce FC-like correlation patterns without necessarily having a biological underpinning. The goal was not to have an abstract converter that simply transforms EEG into an fMRI modality such as time series. Rather, EEG source activity serves as an approximation of ongoing subject-specific synaptic currents and parameter fitting is performed to tune the model to optimally explain empirical fMRI time series. In contrast to a simple 'converter', our biophysical model is able to additionally capture other features of functional brain data not used for model fitting. We show that in fact the parameter space converges for different metrics of brain activity toward a single optimal subspace indicating validity of our model. In our approach, both functional datasets, EEG and fMRI, are fused within the framework of a biophysically grounded and structurally constrained model in order to optimally approximate the underlying (but unobservable) behavior and parameters of the real system. Models, by definition, omit features of the modeled system for the sake of simplicity, generality and efficiency. Adding degrees of freedom renders parameter spaces increasingly intractable and increases the risk of over-fitting. Injection of source activity is a way to systematically probe sufficiently abstract neural systems while maintaining biologically realistic behavior. Thereby, the approach aims to balance a level of abstraction that is sufficient to provide relevant insights, with being detailed enough to guide subsequent empirical study. It is not the goal of this approach to attain the highest possible fit between different imaging modalities at the cost of biological plausibility, which would be the case for abstract statistical models that do not relate to biological entities and therefore preclude the inference of neurophysiological knowledge. Here, imperfect reproduction of neural activity directly points to deficits in our understanding and conceptualization of large-scale brain structure and function, which to iteratively improve is the goal of this approach. We note that our comparison of prediction qualities of the hybrid model and the three control scenarios is not a result in the sense of

formal model comparison where goodness of fit is assessed in light of model complexity. Rather, the informal comparison serves to better assess the hybrid model's prediction quality in relation to the original model and the α-regressor. Although it was a priori clearly unlikely that the noise-driven model or the injection of time permuted EEG would correlate with the empirical time series, these controls serve to exclude that hybrid model correlations were obtained by a trivial mechanism potentially also present in noise models. Furthermore, to test whether it is the specific temporal sequence of time points in the injected activity that enabled fMRI prediction, we simulated the hybrid model's response to permuted input time series. More importantly, these correlations enable us to show that although noise and permuted input do not produce noteworthy time series correlations, like the α-regressor, they nevertheless predict FC, while the hybrid model predicts both, time series and FC.Although the α-regressor makes noteworthy fMRI time series predictions, it yields low correlations with FC and, importantly, it is unable to predict the electric neuronal phenomena that have been reproduced with the hybrid model as it is not based on state variables that correspond to biological entities like the hybrid model. Hence, if during formal model comparison model complexity is penalized without accounting for the accuracy of the model to predict diverse data sets that originate from different modalities and that involve different kinds of metrics (as the hybrid model does), then it is likely that the α-regressor is favoured, because it relies on zero free parameters while achieving similar time series prediction, despite the fact that it clearly has less power to concurrently explain the different sorts of neuronal phenomena explained by the hybrid model. In order to better estimate the relative quality of this kind of models, we are working on a theoretical framework that extends existing Bayesian system identification frameworks (*Friston et al., 2003*) to account for the concurrent prediction of the dynamics of different biological phenomena, data sets and metrics which goes beyond the scope of this study and shall be the subject of an additional publication.

The idea of the hybrid approach is to test how biophysically based and structurally constrained models respond to biologically plausible synaptic input currents, comparable to in vivo or in vitro electrophysiology current injection experiments. However, it must be noted that the hybrid model is clearly limited by the fact that it is not an autonomous (self-contained) model of the brain, but depends on externally injected activity. Furthermore, EEG-based approximation of local EPSCs is limited by the coarse spatial resolution of EEG and the inability to disentangle local EPSCs from other currents that contribute to EEG as all currents in the brain superimpose at any given point in space to generate a potential at that location (*Buzsáki et al., 2012*). This limitation would become apparent when the hybrid model is coupled with a forward model to predict EEG on the basis of the entire sum of input currents (*Equations 1 and 2*). However, when predicting EEG on the basis of local EPSCs only by application of the forward model, this would again yield the original EEG. Notably, EEG source activity can only be viewed as an approximation of EPSCs and it is unclear how EEG exactly relates to EPSCs, that is, to which extend this approximation reflects biological reality. Although theoretical considerations suggest that excitatory postsynaptic potentials dominate current source density (CSD) amplitudes (*Mitzdorf, 1985*), empirical observations repeatedly showed exceptions to this proposition. For example, CSD profiles of neuronal oscillations that were entrained to rhythmic stimulus streams showed a temporal alternation of states dominated by net ensemble depolarization and hyperpolarization, indicating the contribution of IPSCs to CSD profiles (*Lakatos et al., 2008*; *Lakatos et al., 2013*). Despite these limitations several empirical phenomena were reproduced and the input injection approach opens up avenues for future research to investigate the neural mechanisms underlying a wide range of different phenomena. For example, an important feature of the α-rhythm is its characteristic bistable jumping between low-power and high-power modes and a 'dwelling' in each state that follows a stretched-exponential (*Freyer et al., 2009a*). This behavior was remarkably closely reproduced by a multistable corticothalamic model that identified the underlying mechanism as a multistable switching between a fixed point and a limit cycle attractor that is driven by noise (*Freyer et al., 2011*). Importantly, the closest reproduction of EEG α-switching in the model of *Freyer et al. (2011)* emerged only when the uncorrelated Gaussian noise term (injected into mean membrane potentials) was replaced by a state-dependent (autoregressive) noise term, which made the injected stochastic fluctuations effectively autocorrelated. This result is interesting in the context of the present study as our simulations identified the switching between high- and low-power modes of the α-rhythm as a potential generative mechanism underlying fMRI resting-state oscillations. Extending from these results, future BNM studies could

systematically investigate the role of autocorrelated compared to Gaussian inputs and their impact on emerging fMRI dynamics (like FC dynamics), especially since inputs like the EEG source activity used in our hybrid model better capture the autocorrelation structure of biological source currents, which are different from white noise (*Haider et al., 2016*; *Okun et al., 2010*).

In line with our results, cellular-level studies indicate that rhythmic GABAergic input from the interneuronal network is associated with E/I balance (*Dehghani et al., 2016*) and α-related firing (*Jensen and Mazaheri, 2010*; *Lorincz et al., 2009*; *Osipova et al., 2008*). However, the identification of an exact physiological mechanism that explains how α-rhythms can produce an inhibitory effect remained elusive (*Jensen and Mazaheri, 2010*; *Klimesch, 2012*). *Mazaheri and Jensen (2010)* suggest that α-related inhibition occurs due to an observed amplitude asymmetry of ongoing oscillations, also termed baseline-shift. Our results suggest, in accordance with the model from *Mazaheri and Jensen (2010)*, that a symmetrically oscillating driving signal in the α-range leads to asymmetric firing rates and synaptic currents, but we extend this scheme with an explicit explanation of the generation of inhibitory pulses from oscillating input currents. Furthermore, our results with artificial α-activity may help to shed new light on the 'gating by inhibition' hypothesis, which posits that information is routed through the brain network by functionally blocking off task-irrelevant pathways and that this inhibition is reflected by α-activity (*Jensen and Mazaheri, 2010*). In agreement with this hypothesis, we found that long-range input decreased during states of high α-power and increased again when α-power decreased, but further studies are required to examine the effect of α-power on long-range communication and its interaction with other frequency bands.

It is unclear to which degree non-neuronal processes affect the fMRI signal, as different physiological signals such as respiration and cardiac pulse rate were shown to be correlated with resting-state oscillations (*Biswal et al., 1996*; *Power et al., 2017*), which raised concerns that RSN oscillations may be unrelated to neuronal information processing, but rather constitute an epiphenomenon (*Birn et al., 2006*; *de Munck et al., 2008*; *Shmueli et al., 2007*; *Yuan et al., 2013*). The interpretation and handling of these signal modulations is therefore hotly debated and they are often considered as artefactual and removed from fMRI studies (*Birn et al., 2006*; *Chang and Glover, 2009*). Importantly, however, low-frequency BOLD fluctuations are also strongly correlated with electrical neural activity, which was shown by studies that analysed fMRI jointly with EEG (*Goldman et al., 2002*; *Laufs et al., 2003*; *Moosmann et al., 2003*), intracortical recordings (*He et al., 2008*; *Logothetis et al., 2001*) or MEG (*Brookes et al., 2011*; *de Pasquale et al., 2010*). Similarly, strong temporal correlations and spatially similar correlation maps of EEG α-power, respiration and BOLD (*Yuan et al., 2013*), as well as of EEG α-power, heart rate variations and BOLD (*de Munck et al., 2008*) suggest that these fMRI fluctuations are not unrelated to neural activity, but may be of neural origin.

Our results extend the current understanding by showing an explicit mechanism for a neural origin of fMRI RSN oscillations that explains a large part of their variance by a chain of neurophysiological interactions. That is, our simulated activity not only reproduces the negative correlation between α-power fluctuations and BOLD signal, but also reveals a mechanism that transforms ongoing α-power fluctuation into fMRI oscillations. The hybrid approach therefore constitutes a multimodal data fusion approach (*Friston, 2009*; *Valdes-Sosa et al., 2009*) that enables the direct characterization of the previously reported temporal correlations between BOLD and EEG signals in terms of the underlying neural activity and explicit forward models. In addition to fMRI time series, the hybrid model also reproduces the spatial topology of fMRI networks, which are not predicted by the α-power regressor. These findings thereby add to accumulating evidence suggesting that RSNs originate from neuronal activity (*Brookes et al., 2011*; *de Pasquale et al., 2010*; *Goldman et al., 2002*; *He et al., 2008*; *Logothetis et al., 2001*; *Mantini et al., 2007*; *Moosmann et al., 2003*) rather than being a purely hemodynamic phenomenon that is only correlated, but not caused by it (*Birn et al., 2006*; *de Munck et al., 2008*; *Shmueli et al., 2007*). The conclusions from these results have important implications for future fMRI studies, as they implicate that low-frequency fMRI oscillations may be attributed to a neural process that has a considerable state-dependent effect on neural information processing as indicated by the large modulations of neuronal firing and synaptic activity. Methods for physiological noise correction might remove variance from fMRI experiments that is related to neuronal activity and may therefore exclude relevant information for the interpretation of fMRI data.

Parameter space exploration shows that structural coupling is critical for fMRI prediction, as prediction quality decreases for sub-optimal global coupling strengths or when global coupling is deactivated altogether (*Figure 3—figure supplement 1*, *Figure 7—figure supplement 2*). In this study, we did not address the effect of coupling time delays, as they were non-essential for the emergence of the described phenomena. Our initial application of the novel hybrid model aimed to study the effect of input injection while minimizing the degrees of freedom of the simulation and the set of parameters to be varied. Further studies are required to determine the effect of coupling delays, as previous studies demonstrated their important role for emerging large-scale dynamics (*Deco et al., 2011*; *Jirsa, 2008*, *2009*). We observed that the prediction quality of resting-state network activation time courses fluctuates over time and is highest during epochs of highest variance of the respective temporal mode. During these, time windows resting-state networks contribute the largest variance to whole-brain fMRI, that is, they are the most active. A possible explanation may be that during states of asynchronous neural activity (i.e. when the variance of RSN temporal modes is low) volume conduction and cancellation of electromagnetic waves decreases the ability of source imaging methods to reconstruct source activity.

It is important to note that the observed processes may not be specific to α-oscillations, but may apply also to other frequencies or non-oscillatory signal components, for example, phase-locked discharge of neurons occurs over a range of frequency bands and is not limited to the α-rhythm (*Buzsaki, 2006*). Furthermore, the α-rhythm, though prominent, is certainly not superior to other rhythms with respect to neuronal computation and cognition (*Fries, 2015*). In fact, it may be best thought of as one of several modes of brain operation, even during the so-called resting-state (*Engel et al., 2013*). Additional empirical and theoretical studies will be needed to address these limitations more comprehensively. Although our analysis revolved around α-oscillations, the hybrid modeling approach is not restricted to α-activity, as the injected EEG source activity was not limited to the α-band. The hybrid modeling approach itself does not set any requirements on the frequency spectrum of the injected source activity. Importantly, our focus on α-rhythms was not 'by construction', but, as outlined in the introduction, emerged from a sequence of analyses that we performed to understand how the hybrid model generated the correlation with empirical fMRI time series. In this regard, it is interesting to note that the time series correlations obtained by the α-regressor and the hybrid model are comparable, which indicates that the α-rhythm was the main driver for the hybrid model's fMRI time series prediction.

Despite the ubiquity of scale invariant dynamics, models that generate power-law distributions are often rather generic and detached from the details of the modeled systems (*Bak et al., 1987*; *Marković and Gros, 2014*). Furthermore, the precise mechanisms that lead to the emergence of fMRI power spectrum power-law scaling or the relationship between brain network interaction and fMRI power-law scaling are unclear (*He, 2011*). Our simulation results indicate that fMRI spectra power-law scaling is due to the observed frequency-dependent amplification of oscillatory activity in networks that contain self-reinforcing feedback excitation together with slow decay of activity. Central to theories on the emergence of criticality is the tuning of a control parameter (e.g. connection strengths) that leads the system to a sharp change in one or more order parameters (e.g. firing rates) when the control parameter is moved over a critical point that marks the boundary of a phase transition. It is important to point out that the existence of power-laws alone does not prove criticality. Rather, criticality requires the existence of a control parameter that can be adjusted to move the system through a phase transition at a critical point (*Beggs and Timme, 2012*). In vivo, in vitro and in silico results show that the dynamical balance between excitation and inhibition was found to be essential to move the system towards or away from criticality, for example, by pharmacologically altering the excitation-inhibition balance in anesthetized rats (*Osorio et al., 2010*), acute slices (*Beggs and Plenz, 2003*) or by changing parameters that control global excitation and inhibition in computational models (*Deco et al., 2014*). However, the exact role played by excitation-inhibition balance is unclear. In line with these results, we found that power-law scaling varied as a function of the relative levels of global excitation and inhibition, further emphasizing the need for a proportional relationship between these control parameters (*Figure 7—figure supplement 2*). Extending from that, our simulation results indicate that E/I balance may cause a tuning of the relative strengths of local and long-range inputs to neural populations that supports constructive interference between the different input currents, which in turn amplifies slower oscillations more than faster oscillation. These results address an open question on whether power-laws in neural networks result from

power-law behavior on the cellular level or from a global network-level process (*Beggs and Timme, 2012*), by giving an explanation for scale-free fMRI power spectra as an emergent property of long-range brain network interaction that does not require small-scale decentralized processes like the constant active retuning of microscopic parameters as proposed in some theories of self-organized criticality (*Bak et al., 1987*; *Hesse and Gross, 2014*). Furthermore, these results explicitly address the effect of input activity, while in vitro and in silico studies have so far focused on systems without or considerably decreased input (*Hesse and Gross, 2014*). The observed co-emergence of spatial long-range correlations (i.e. functional connectivity networks) and power-law scaling may point to a unifying explanation within the theory of self-organized criticality, as previously proposed by others (*Linkenkaer-Hansen et al., 2001*). Note that we have used the hybrid model not simply to establish the prevalence of scale invariant dynamics, but to use the power law scaling in a quantitative sense to understand the mechanisms leading to particular power law exponents; for example, the importance of extrinsic (between node) connections in explaining the differences between power law scaling at the electrophysiological and haemodynamic level. This is an important point because scale-free behavior per se would be difficult to avoid in simulations of this sort.

A wide range of disorders like autism, schizophrenia, intellectual disabilities, Alzheimer's disease, multiple sclerosis or epilepsy have been linked to disruption of E/I balance (*Marín, 2012*) and altered structural and functional network connectivity (*Stam, 2014*). The presented modelling approach may therefore play a key role for identifying the precise mechanisms underlying the pathophysiology of different disorders and assist in developing novel therapies that restore altered E/I balance or brain connectivity, for example, by identifying the targets for neural stimulation therapies or by guiding individually customized therapy. The ability of the hybrid model to infer precise neurophysiological mechanisms that give rise to empirical phenomena and to link the involved mechanisms and signal patterns across different scales and neuroimaging modalities makes it a potentially valuable tool for neuroscience research.

# Materials and methods

## Computational model

The model used in this study is based on the large-scale dynamical mean field model used by Deco and colleagues (*Deco et al., 2014*; *Wong and Wang, 2006*). Brain activity is modeled as the network interaction of local population models that represent cortical areas. Cortical regions are modelled by interconnected excitatory and inhibitory neural mass models. In contrast to the original model, excitatory connections were replaced by injected EEG source activity. The dynamic mean field model faithfully approximates the time evolution of average synaptic activities and firing rates of a network of spiking neurons by a system of coupled non-linear differential equations for each node $i$:

$$I_i^{(E)} = W_E I_0 + G \sum_j C_{ij} S_j^{(E)} - J_i S_i^{(I)} + w_{BG}^{(E)} I_{BG} \tag{1}$$

$$I_i^{(I)} = W_I I_0 - S_i^{(I)} + w_{BG}^{(I)} I_{BG} \tag{2}$$

$$r_i^{(E)} = \frac{a_E I_i^{(E)} - b_E}{1 - exp\left(-d_E\left(a_E I_i^{(E)} - b_E\right)\right)} \tag{3}$$

$$r_i^{(I)} = \frac{a_I I_i^{(I)} - b_I}{1 - exp\left(-d_I\left(a_I I_i^{(I)} - b_I\right)\right)} \tag{4}$$

$$\frac{dS_i^{(E)}(t)}{dt} = -\frac{S_i^{(E)}}{\tau_E} + \left(1 - S_i^{(E)}\right)\gamma_E r_i^{(E)} \tag{5}$$

$$\frac{dS_i^{(I)}(t)}{dt} = -\frac{S_i^{(I)}}{\tau_I} + \gamma_I r_i^{(I)} \qquad (6)$$

Here, $r_i^{(E,I)}$ denotes the population firing rate of the excitatory ($E$) and inhibitory ($I$) population of brain area $i$. $S_i^{(E,I)}$ identifies the average excitatory or inhibitory synaptic gating variables of each brain area, while their input currents are given by $I_i^{(E,I)}$. In contrast to the model used by *Deco et al. (2014)* that has recurrent and feedforward excitatory coupling, we approximate excitatory postsynaptic currents $I_{BG}$ using region-wise aggregated EEG source activity that is added to the sum of input currents $I_i^{(E,I)}$. This approach is based on intracortical recordings that suggest that EPSCs are non-random, but strongly correlated with electric fields in their vicinity, while IPSCs are anticorrelated with EPSCs (*Haider et al., 2016*). The weight parameters $\omega_{BG}^{(E,I)}$ rescale the z-score normalized EEG source activity independently for excitatory and inhibitory populations. $G$ denotes the long-range coupling strength scaling factor that rescales the structural connectivity matrix $C_{ij}$ that denotes the strength of interaction for each region pair $i$ and $j$. All three scaling parameters are estimated by fitting simulation results to empirical fMRI data by exhaustive search. Initially, parameter space ($n$-dimensional real space with $n$ being the number of optimized parameters) was constrained such that the strength of inhibition was larger than the strength of excitation, satisfying a biological constraint. Furthermore, for each tested parameter set (containing the three scaling parameters mentioned above), the region-wise parameters $J_i$ that describe the strength of the local feedback inhibitory synaptic coupling for each area $i$ (expressed in nA) are fitted with the algorithm described below such that the average firing rate of each excitatory population in the model was close to 3.06 Hz (i.e. the cost function for tuning parameters $J_i$ was solely based on average firing rates and not on prediction quality). The overall effective external input $I_0 = 0.382$ nA is scaled by $W_E$ and $W_I$, for the excitatory and inhibitory pools, respectively. $r_i^{(E,I)}$ denotes the neuronal input-output functions (f-I curves) of the excitatory and inhibitory pools, respectively. All parameters except those that are tuned during parameter estimation are set as in *Deco et al. (2014)*. Please refer to *Table 1* for a specification of state variables and parameters. BOLD activity was simulated on the basis of the excitatory synaptic activity $S^{(E)}$ using the Balloon-Windkessel hemodynamic model (*Friston et al., 2003*), which is a dynamical model that describes the transduction of neuronal activity into perfusion changes and the coupling of perfusion to BOLD signal. The model is based on the assumption that the BOLD signal is a static non-linear function of the normalized total deoxyhemoglobin voxel content, normalized

**Table 1.** State variables and parameters of the hybrid brain network model.

| Quantity | Value | Description |
|---|---|---|
| *State variables* | | |
| $r_i(E,I)$ | | Population firing rate of the excitatory ($E$) or inhibitory ($I$) population in brain area $i$ |
| $S_i(E,I)$ | | Average synaptic gating |
| $I_i(E,I)$ | | Sums of all input currents |
| $I_{BG}$ | | EEG-derived input currents |
| *Parameters* | | |
| $w_+$ | 1.4 | Local excitatory recurrence |
| $C_{ij}$ | Obtained from diffusion tractography | Structural connectivity matrix |
| $\gamma_E, \gamma_I$ | $6.41 \times 10^{-4}$, $1.0 \times 10^{-3}$ | Kinetic parameters |
| $a_E, b_E, d_E, \tau_E, W_E$ | 310 (nC$^{-1}$), 125 (Hz), 0.16 (s), 100 (ms), 1 | Excitatory gating variables |
| $a_I, b_I, d_I, \tau_I, W_I$ | 615 (nC$^{-1}$), 177 (Hz), 0.087 (s), 10 (ms), 0.7 | Inhibitory gating variables |
| $J_{NMDA}$ | 0.15 (nA) | Excitatory synaptic coupling |
| $J_I$ | Obtained by FIC heuristic (nA) | Feedback inhibitory synaptic coupling |
| $I_0$ | 0.382 (nA) | overall effective external input |
| $G$ | Obtained from model fitting | Global coupling scaling factor |
| $w_{BG}^{(E)}, w_{BG}^{(I)}$ | Obtained from model fitting | Weights for scaling EEG-derived input currents |

DOI: https://doi.org/10.7554/eLife.28927.015

venous volume, resting net oxygen extraction fraction by the capillary bed, and resting blood volume fraction. Please refer to *Deco et al. (2013)* for the specific set of Ballon-Windkessel model equations that we used in this study.

## Parameter optimization

For each brain network model, three parameters were varied to maximize the fit between empirical and simulated fMRI: the scaling of excitatory white-matter coupling and the strengths of the inputs injected into excitatory and inhibitory populations (please refer to *Table 2* for an overview over the obtained parameter values). Following in vivo observations (*Xue et al., 2014*), we ensured that at excitatory populations EPSC amplitudes are smaller than IPSC amplitudes by constraining the range of values for the ratio $\omega_{BG}^{(I)} / \omega_{BG}^{(E)}$ between 5 and 200, which we found through initial pilot simulations. Note that the ratio $\omega_{BG}^{(I)} / \omega_{BG}^{(E)}$ is not identical to the amplitude ratio of IPSCs vs. EPSCs, but depends also on the specific settings of all other varied parameters. For example, a large ratio $\omega_{BG}^{(I)} / \omega_{BG}^{(E)}$ can still lead to a small ratio of IPSCs vs. EPSCs amplitudes if the local feedback inhibition parameter $J_i$ is small. Apart from these initial pilot simulations to restrict the ratio of postsynaptic currents to a biologically plausible range, the specific combination of all varied parameters was exclusively found through fitting simulated to empirical fMRI time series under the constraint of plausible firing rates. That is, besides tuning these three global parameters using the sole optimization criterion of maximizing the fit between simulated and empirical fMRI time series, we adjusted local inhibitory coupling strengths in order to obtain biologically plausible firing rates in excitatory populations. For this second form of tuning, termed feedback inhibition control (FIC), average population firing rates were the sole optimization criterion, without any consideration of prediction quality, which was only dependent on the three global parameters. FIC modulates the strengths of inhibitory connections that is required to compensate for excess or lack of excitation resulting from the large variability in white-matter coupling strengths obtained by MRI tractography, which is a prerequisite to obtain plausible ranges of population activity that is relevant for some results (*Figure 5* and *Figure 6*). Prediction quality was measured as the average correlation coefficient between all simulated and empirical region-wise fMRI time series of a complete cortical parcellation over 20.7 min length (TR = 1.94 s, 640 data points) thereby quantifying the ability of the model to predict the activity of 68 parcellated cortical regions. Accounting for the large-scale nature of fMRI resting-state networks, the chosen parcellation size provides a parsimonious trade-off between model complexity and the desired level of explanation. What this parcellation may lack in spatial detail, it gains in providing a

**Table 2.** Parameters of the 15 hybrid brain network models obtained by parameter tuning.

| G | $w_{BG}^{(I)}$ | $w_{BG}^{(I)}/w_{BG}^{(E)}$ |
|---|---|---|
| 0.12 | 0.13 | 5 |
| 0.1 | 0.03 | 5 |
| 0.14 | 0.08 | 50 |
| 0.09 | 0.03 | 10 |
| 0.13 | 0.15 | 5 |
| 0.44 | 0.03 | 25 |
| 0.15 | 0.05 | 20 |
| 0.09 | 0.12 | 5 |
| 0.48 | 0.04 | 125 |
| 0.2 | 0.15 | 10 |
| 0.11 | 0.12 | 10 |
| 0.12 | 0.02 | 5 |
| 0.25 | 0.04 | 5 |
| 0.44 | 0.07 | 200 |
| 0.09 | 0.04 | 5 |

DOI: https://doi.org/10.7554/eLife.28927.016

full-brain coverage that can reliably reproduce ubiquitous large-scale features of empirical data, which we further present below. To exclude overfitting and limited generalizability, a five-fold cross-validation scheme was performed on the hybrid model simulation results. Therefore, the data was randomly divided into two subsets: 80% as training subset and 20% as testing subset. Prediction quality was estimated using the training set, before trained models were asked to predict the testing set. Resulting prediction quality was compared between training and test data set and between test data set and the data obtained from fitting the full time series. Furthermore, despite the large range of possible parameters, the search converged to a global maximum (*Figure 3—figure supplement 1*). Therefore, we ensured that when the model has been fit to a subset of empirical data, that it was able to generalize to new or unseen data. In contrast to model selection approaches, where the predictive power of different models and their complexity are compared against each other, we here use only a single type of model.

## Feedback inhibition control

The excitatory populations of isolated nodes of the original model described in *Deco et al. (2014)* have an average firing rate of 3.06 Hz. That is, without long-range coupling $G \sum_j C_{ij} S_j^{(E)}$ and without injected activity $w_{BG}^{(E)} I_{BG}$ and $w_{BG}^{(I)} I_{BG}$ (cf. *Equations 1 and 2*), the used excitatory populations have an average firing rate of 3.06 Hz. This value conforms to the empirically measured Poisson-like cortical in vivo activity of ~3 Hz (*Softky and Koch, 1993*; *Wilson et al., 1994*) and results from the dynamic mean field approximation of the average ensemble behaviour of a large-scale spiking neuron model used in *Deco et al. (2014)*. In contrast to isolated nodes, the firing rate of coupled nodes change in dependence of the employed structural connectivity matrix and the injected input. To compensate for a resulting excess or lack of excitation, a local regulation mechanism, called feedback inhibition control (FIC), was used. The approach was previously successfully used to significantly improve FC prediction as well as for increasing the dynamical repertoire of evoked activity and the accuracy of external stimulus encoding (*Deco et al., 2014*). Despite the mentioned advantages of FIC tuning, it has the disadvantage of increasing the number of open parameters of the model. To prove that prediction quality is not due to FIC, but solely due to the three global parameters and to exclude concerns about over-parameterization or that FIC may be a potentially necessary condition for the emergence of scale-freeness, we devised a control model that did not implement FIC, but used a single global parameter for inhibitory coupling strength. Instead of tuning the 68 individual local coupling weights individually, only a single global value for all inhibitory coupling weights $J_i$ was varied. We compared the effect of FIC on time series prediction quality and found no significant difference in prediction quality to simulations that used only a single value for all local coupling weights $J_i$ per subject (one-tailed Wilcoxon rank sum test, p=0.36, z = −0.37, Cliffs's delta d = −0.15). In contrast to simulations that are driven by noise (*Deco et al., 2014*), FIC parameters for injected input must be estimated for the entire simulated time series, since the non-stationarity of stimulation time series leads to considerable fluctuations of firing rates. Therefore, we developed a local greedy search algorithm for fast FIC parameter estimation based on the algorithm in *Deco et al. (2014)*. To exert FIC, local inhibitory synaptic strength is iteratively adjusted until all excitatory populations attained a firing rate close to the desired mean firing rates for the entire ~20 min of activity. During each iteration, the algorithm performs a simulation of the entire time series. Then, it computes the mean firing activity over the entire time series for each excitatory population and adapts $J_i$ values accordingly, that is, it increases local $J_i$ values if the average firing rate over all excitatory populations during the $k$-th iteration $\hat{r}_k$ is larger than 3.06 Hz and vice versa. In order to reduce the number of iterations the value by which $J_i$ is changed is, in contrast to the algorithm by *Deco et al. (2014)*, dynamically adapted in dependence of the firing rate obtained during the current iteration

$$J_i^{k+1} = J_i^k + (\hat{r}_k - 3.06)\tau_k \tag{7}$$

where $J_i^k$ denotes the value of feedback inhibition strength of node $i$ and $\tau_k$ denotes the adaptive tuning factor during the $k$-th iteration. In the first iteration, all $J_i$ values are initialized with one and $\tau_k$ is initialized with 0.005. The adaptive tuning factor is dynamically changed during each iteration based on the result of the previous iteration:

$$\tau_{k+1} = \left( \sum_i (J_i^{k-1} - J^k) \right) / (\hat{r}^{k-1} - \hat{r}^k). \tag{8}$$

For the case that the result did not improve during the current iteration, that is,

$$|\hat{r} - 3.06| \geq |\hat{r}^{k-1} - 3.06|, \tag{9}$$

the adaptive tuning factor is decreased by multiplying it with 0.5 and the algorithm continues with the next iteration. After 12 iterations, all $J_i$ values are set to the values they had during the iteration $k$ where $|\hat{r}^k - 3.06|$ was minimal.

## MRI preprocessing

Structural and functional connectomes from 15 healthy human subjects (age range: 18–31 years, eight female) were extracted from full data sets (diffusion-weighted MRI, T1-weighted MRI, EEG-fMRI) using a local installation of a pipeline for automatic processing of functional and diffusion-weighted MRI data (*Schirner et al., 2015*). From a local database of 49 subjects (age range 18–80 years, 30 female) that was acquired for a previous study (*Schirner et al., 2015*), we selected the 15 youngest subjects that fulfilled highest EEG quality standards after applying MR artefact correction routines. EEG quality was assessed by standards that were defined prior to the experimental design and that are routinely used in the field (*Becker et al., 2011*; *Freyer et al., 2009b*; *Ritter et al., 2010*; *Ritter et al., 2007*): occurrence of spikes in frequencies > 20 Hz in power spectral densities, excessive head motion and cardio-ballistic artefacts. Research was performed in compliance with the Code of Ethics of the World Medical Association (Declaration of Helsinki). Written informed consent was provided by all subjects with an understanding of the study prior to data collection, and was approved by the local ethics committee in accordance with the institutional guidelines at Charité Hospital Berlin. Subjects with a self-reported history of neurological, cognitive, or psychiatric conditions were excluded from the experiment. Structural (T1-weighted high-resolution three-dimensional MP-RAGE sequence; TR = 1,900 ms, TE = 2.52 ms, TI = 900 ms, flip angle = 9°, field of view (FOV) = 256 mm x 256 mm x 192 mm, 256 × 256 × 192 Matrix, 1.0 mm isotropic voxel resolution), diffusion-weighted (T2-weighted sequence; TR = 7500 ms, TE = 86 ms, FOV = 192 mm x 192 mm, 96 × 96 Matrix, 61 slices, 2.3 mm isotropic voxel resolution, 64 diffusion directions), and fMRI data (two-dimensional T2-weighted gradient echo planar imaging blood oxygen level-dependent contrast sequence; TR = 1,940 ms, TE = 30 ms, flip angle = 78°, FOV = 192 mm x 192 mm, 3 mm x 3 mm voxel resolution, 3 mm slice thickness, 64 × 64 matrix, 33 slices, 0.51 ms echo spacing, 668 TRs, 7 initial images were acquired and discarded to allow magnetization to reach equilibrium; eyes-closed resting-state) were acquired on a 12-channel Siemens 3 Tesla Trio MRI scanner at the Berlin Center for Advanced Neuroimaging, Berlin, Germany. Extracted structural connectivity matrices intend to give an aggregated representation of the strengths of interaction between regions as mediated by white matter fiber tracts. As in the original model by *Deco et al. (2014)*, conduction delays were neglected in this study as they were non-essential for the described features. Strength matrices $C_{ij}$ were divided by their respective maximum value for normalization. In short, the pipeline proceeds as follows: for each subject a three-dimensional high-resolution T1-weighted image image was used to divide cortical gray matter into 68 regions according to the Desikan-Killiany atlas using FreeSurfer's (*Fischl, 2012*) automatic anatomical segmentation and registered to diffusion data. The gyral-based brain parcellation is generated by an automated probabilistic labeling algorithm that has been shown to achieve a high level of anatomical accuracy for identification of regions while accounting for a wide range of inter-subject anatomical variability (*Desikan et al., 2006*). The atlas was successfully used in previous modelling studies and provided highly significant structure-function relationships (*Honey et al., 2009*; *Ritter et al., 2013*; *Schirner et al., 2015*). Details on diffusion-weighted and fMRI preprocessing can be found in Schirner et al. (*Schirner et al., 2015*) Briefly, probabilistic white matter tractography and track aggregation between each region-pair was performed as implemented in the automatic pipeline and the implemented distinct connection metric extracted. This metric weights the raw track count between two regions according to the minimum of the gray matter/white matter interface areas of both regions used to connect these regions in distinction to other metrics that use the unweighted raw track count, which was shown to be biased by subject-specific anatomical features (see *Schirner et al. (2015)* for a discussion). After preprocessing, the cortical parcellation mask was registered to fMRI resting-state data of subjects and average fMRI signals for

each region were extracted. The first five images of each scanning run were discarded to allow the MRI signal to reach steady state. To identify RSN activity a spatial Group ICA decomposition was performed for the fMRI data of all subjects using FSL MELODIC (*Beckmann and Smith, 2004*) (MELODIC v4.0; FMRIB Oxford University, UK) with the following parameters: high pass filter cut off: 100 s, MCFLIRT motion correction, BET brain extraction, spatial smoothing 5 mm FWHM, normalization to MNI152, temporal concatenation, dimensionality restriction to 30 output components. ICs that correspond to RSNs were automatically identified by spatial correlation with the 9 out of the 10 well-matched pairs of networks of the 29,671-subject BrainMap activation database as described in *Smith et al. (2009)* (excluding the cerebellum network). All image processing were performed in the native subject space of the different modalities and the brain atlas was transformed from T1-space of the subject into the respective spaces of the different modalities.

## EEG preprocessing

Details of EEG preprocessing are described in supplementary material of Schirner et al. (*Schirner et al., 2015*). First, to account for slow drifts in EEG channels and to improve template construction during subsequent MR imaging acquisition artefact (IAA) correction all channels were high-pass filtered at 1.0 Hz (standard FIR filter). IAA correction was performed using Analyser 2.0 (v2.0.2.5859, Brain Products, Gilching, Germany). The onset of each MRI scan interval was detected using a gradient trigger level of 300 µV/ms. Incorrectly detected markers, for example due to shimming events or heavy movement, were manually rejected. To assure the correct detection of the resulting scan start markers each inter-scan interval was controlled for its precise length of 1940 ms (TR). For each channel, a template of the IAA was computed using a sliding average approach (window length: 11 intervals) and subsequently subtracted from each scan interval. For further processing, the data were down sampled to 200 Hz, imported to EEGLAB and low-pass filtered at 60 Hz. ECG traces were used to detect and mark each instance of the QRS complex in order to identify ballistocardiogram (BCG) artefacts. The reasonable position and spacing of those ECG markers was controlled by visual inspection and corrected if necessary. To correct for BCG and artefacts induced by muscle activity, especially movement of the eyes, a temporal ICA was computed using the extended Infomax algorithm as implemented in EEGLAB. To identify independent components (ICs) that contain BCG artefacts the topography plot, activation time series, power spectra and heartbeat triggered average potentials of the resulting ICs were used as indication. Based on established characteristics, all components representing the BCG were identified and rejected, that is, the components were excluded from back-projection. The remaining artificial, non-BCG components, accounting for primarily movement events especially eye movement, were identified by their localization, activation, power spectral properties and ERPs. Detailed descriptions of EEG and fMRI preprocessing have been published elsewhere (*Becker et al., 2011*; *Freyer et al., 2009a*; *Ritter et al., 2010*; *Ritter et al., 2007*).

## Biologically based model input

EEG source imaging was performed with the freely available MATLAB toolbox Brainstorm using default settings and standard procedure for resting-state EEG data as described in the software documentation (*Tadel et al., 2011*). Source space models were based on the individual cortical mesh triangulations as extracted by FreeSurfer from each subject's T1-weighted MRI data and downsampled by Brainstorm. From the same MRI data, head surface triangulations were computed by Brainstorm. Standard positions of the used EEG caps (Easy-cap; 64 channels, MR compatible) were aligned by the fiducial points used in Brainstorm and projected onto the nearest point of the head surface. Forward models are based on Boundary Element Method head models computed using the open-source software OpenMEEG and 15002 perpendicular dipole generator models located at the vertices of the cortical surface triangulation. The sLORETA inverse solution was used to estimate the distributed neuronal current density underlying the measured sensor data since it has zero localization error (*Pascual-Marqui, 2002*). EEG data were low-pass filtered at 30 Hz and imported into Brainstorm. There, the epochs before the first and after the last fMRI scan were discarded and the EEG signal was time-locked to fMRI scan start markers. Using brainstorm routines, EEG data were projected onto the cortical surface using the obtained inversion kernel and averaged according to the Desikan-Killiany parcellation that was also used for the extraction of structural and functional

connectomes and region-averaged fMRI signals. The resulting 68 region-wise source time series were imported to MATLAB, z-score normalized and upsampled to 1000 Hz using spline interpolation as implemented by the Octave function *interp1*. To enable efficient simulations, the sampling rate of the injected activity was ten times lower than model sampling rate. Hence, during simulation identical values have been injected during each sequence of 10 integration steps.

## Simulation and analysis

Simulations were performed with a highly optimized C implementation of the previously described model on the JURECA supercomputer at the Juelich Supercomputing Center. Simulation and analyses code and used data is open source and available from online repositories (*Schirner et al., 2017a*, see 'Data and code availability'). An exhaustive brute-force parameter space scan using 3888 combinations of the parameters $G$ and $\omega_{BG}^{(E,I)}$ was performed for each subject. Each of these combinations was computed 12 times to iteratively tune $J_i$ values. As control setup, further simulations were performed with random permutations of the input time series. Therefore, the individual time points of each source activity time series were randomly permuted (individually for each region and subject) using the Octave function *randperm()* and injected into simulations using all parameter combinations that were previously used. As an additional control situation the original dynamic mean field model as described in *Deco et al. (2014)* was simulated for the 15 SCs. Here, the parameters $G$ and $J_{NMDA}$ were varied and FIC tuning was performed using the same algorithm as used for the source activity injection model. The simulation and FIC optimization process was identical for all three models. The length of the simulated time series for each subject was 21.6 min. Simulations were performed at a model sampling rate of 10,000 Hz. BOLD time series were computed for every $10^{th}$ time step of excitatory synaptic gating activity using the Balloon-Windkessel model (*Friston et al., 2003*). Since the Balloon-Windkessel model acts like a low-pass filter that attenuates frequencies above ~0.15 Hz (*Robinson et al., 2006*), additional low-pass filtering was unnecessary for downsampling of simulated fMRI time series. Hence, from the resulting time series every 1940th step was stored in order to obtain a sampling rate of simulated fMRI that conforms to the empirical fMRI TR of 1.94 s. The first 11 scans (21.34 s) of activity were discarded to allow model activity and simulated fMRI signal to stabilize. For each subject and modelling approach the simulation result that yielded the highest average correlation between all 68 empirical and simulated region time series for all tested parameters was used for all analyses. To ensure region-specificity of simulation results only corresponding simulated and empirical region time series were correlated in the case of raw fMRI, respectively, for resting-state networks only simulated regions that overlap with the spatial activation pattern of the respective network were used for estimating prediction quality. Specifically, for RSN analysis, only those regions were compared with the temporal modes of RSNs that had a spatial overlap of at least 40% of all voxels belonging to the respective region. To assess time-varying prediction quality, a correlation analysis was performed in which a window with a length of 100 scans (194 s) was slid over the 68 pairs of empirical and simulated time series and the average correlation over all 68 regions was computed for each window. For the estimation of signal correlation, the computation of entries of FC matrices and as a measure of similarity of FC matrices Pearson's linear correlation coefficient was used. FC matrices were compared by stacking all elements below the main diagonal into vectors and computing the correlation coefficient of these vectors. Short-epoch FC prediction quality was estimated by computing the mean correlation obtained for all window-wise correlations of a sliding window analysis of empirical and simulated time series (window-size: 100 scans = 194 s).

To ensure scale-freeness of empirical and simulated signals, region time series were tested using rigorous model selection criteria; on average 79% of all 1020 region-wise time series (15 subjects x 68 regions) for the seven analyzed signal types (empirical fMRI, simulated fMRI, simulated fMRI without global coupling, simulated fMRI without FIC, simulated fMRI without FIC and without global coupling, $\alpha$-power, $\alpha$-regressor) tested as scale-free; for every signal type every subject had at least five regions to test as scale-free. PSDs were computed using the Welch method as implemented in Octave, normalized by their total power and averaged. Resulting average power spectra were fitted with a power-law function $f(x)=ax^{\beta}$ using least-squares estimation in the frequency range 0.01 Hz and 0.17 Hz which is identical to the range for which the test for scale invariance was performed. Frequencies below were excluded in order to reduce the impact of low-frequency signal confounds and scanner drift, frequencies above that limit were excluded to avoid aliasing artefacts in higher

frequency ranges (TR = 1.94 s, hence Nyquist frequency is around 0.25 Hz). In order to compare the scale invariance of our empirical fMRI data with results from previous publications (*He, 2011*), we also computed power spectra in a range that only included frequencies < 0.1 Hz.

In order to adequately determine the existence of scale invariance we applied rigorous model selection to every time series to identify power-law scaling and excluded all time series from analyses that were described better by a model other than a power-law. Nevertheless, we compared the obtained results from this strict regime with results obtained when all time series were included and found them to be qualitatively identical. To test for the existence of scale invariance we used a method that combines a modified version of the well-established detrended fluctuation analysis (DFA) with Bayesian model comparison (*Ton and Daffertshofer, 2016*). DFA is, in contrast to PSD analyses, robust to both stationary and nonstationary data in the presence of confounding (weakly non-linear) trends. It is important to note, that a simple linear fit of the detrended fluctuation curve without proper comparison of the obtained goodness of fit with that of other models would entirely ignore alternative representations of the data different than a power law. For quantification of the goodness of fit with simple regression its corresponding coefficient of determination, $R^2$, is ill-suited as it measures only the strength of a linear relationship and is inadequate for nonlinear regression (*Ton and Daffertshofer, 2016*). It is important to note that with this method the assessment of power-law scaling is based on maximum likelihood estimation, which overcomes the limitations of a minimal least-squares estimate obtained from linear regression in the conventional DFA approach. Details of the used method are described elsewhere (*Ton and Daffertshofer, 2016*). For the different signals the majority of time series were tested as being scale free: 83% for empirical fMRI, 69% for simulated fMRI, 71% for simulated fMRI with deactivated FIC, 83% for simulated fMRI with deactivated global coupling, 86% for simulated fMRI with deactivated global coupling and FIC, 90% for α-power and 70% for the α-regressor.

To compute grand average waveforms, state-variables were averaged over all 15 subjects and 68 regions (N = 1020 region time series) time-locked to the zero crossing of the α-amplitude, which was obtained by band-pass filtering source activity time series between 8 and 12 Hz; to obtain sharp average waveforms, all α-cycle epochs with a cycle length between 95 and 105 ms were used (N = 4,137,994 α-cycle). For computing ongoing α-power time courses, instantaneous power time series were computed by taking the absolute value of the analytical signal (obtained by the Hilbert transform) of band-pass filtered source activity in the 8–10 Hz frequency range; the first and last ~50 s were discarded to control for edge effects. To compute the α-regressor, power time series were convolved with the canonical hemodynamic response function, downsampled to fMRI sampling rate and shifted relative to fMRI time series to account for the lag of hemodynamic response. The highest negative average correlation over all 68 region-pairs obtained within a range of ±3 scans shift was used for comparison with simulation results.

## Statistical analyses

All statistical analyses were performed using MATLAB (The MathWorks, Inc., Natick, Massachusetts, United States). Data are represented as box-and-whisker plots. As normality was not achieved for the majority of data sets (assessed by Lilliefors test at significance level of 0.05), differences between groups were compared by non-parametric statistical tests, using either two-tailed Wilcoxon rank sum test or, in case of directional prediction, one-tailed Wilcoxon rank sum test; a value $p < 0.05$ was considered significant.

## Data and code availability

Brain network models are implemented in the open source neuroinformatics platform The Virtual Brain (*Ritter et al., 2013*; *Sanz-Leon et al., 2015*, *Sanz Leon et al., 2013*) that can be downloaded from thevirtualbrain.org. Code and data that support the findings of this study can be obtained from https://github.com/BrainModes/The-Hybrid-Virtual-Brain (*Schirner et al., 2017b*; copy archived at https://github.com/elifesciences-publications/The-Hybrid-Virtual-Brain) and https://osf.io/mndt8/ (*Schirner et al., 2017a*).

## Acknowledgements

The authors gratefully acknowledge the computing time granted by the John von Neumann Institute for Computing (NIC) provided on the supercomputer JURECA (*Krause and Thörnig, 2016*) at Jülich Supercomputing Centre (www.fz-juelich.de, Grant NIC#8344 and NIC#10276 to PR). The authors acknowledge the support of the James S McDonnell Foundation (Brain Network Recovery Group JSMF22002082) to ARM, VJ, GD, and PR, and funding granted by the German Ministry of Education and Research (US-German Collaboration in Computational Neuroscience 01GQ1504A, Bernstein Focus State Dependencies of Learning 01GQ0971-5, the Max-Planck Society Minerva Program), the European Union Horizon2020 (ERC Consolidator Grant BrainModes 683049), Stiftung Charité/Private Exzellenzinitiative Johanna Quandt and Berlin Institute of Health (BIH Johanna Quandt Professorship for Brain Simulation) to PR. This publication is part of a project that has received funding from the European Union's Horizon 2020 research and innovation programme under grant agreement No BrainModes 683049 (ERC Consolidator grant to PR). The authors thank Olaf Sporns, Jochen Braun and Andreas Daffertshofer for their helpful comments on the manuscript. The authors declare no competing financial interests.

## Additional information

### Funding

| Funder | Grant reference number | Author |
|---|---|---|
| James S. McDonnell Foundation | Brain Network Recovery Group JSMF22002082 | Anthony Randal McIntosh Viktor Jirsa Gustavo Deco Petra Ritter |
| Horizon 2020 | Research and Innovation Programme (720270) | Viktor Jirsa Gustavo Deco |
| Bundesministerium für Bildung und Forschung | Bernstein Focus State Dependencies of Learning 01GQ0971-5 | Petra Ritter |
| Horizon 2020 | ERC Consolidator Grant BrainModes 683049 | Petra Ritter |
| Bundesministerium für Bildung und Forschung | US-German Collaboration in Computational Neuroscience 01GQ1504A | Petra Ritter |
| Bundesministerium für Bildung und Forschung | Max-Planck Society | Petra Ritter |
| John von Neumann Institute for Computing at Jülich Supercomputing Centre | Grant NIC#8344 | Petra Ritter |
| Stiftung Charité/Private Exzellenzinitiative Johanna Quandt and Berlin Institute of Health | | Petra Ritter |
| John von Neumann Institute for Computing at Jülich Supercomputing Centre | Grant NIC#10276 | Petra Ritter |

The funders had no role in study design, data collection and interpretation, or the decision to submit the work for publication.

### Author contributions

Michael Schirner, Conceptualization, Data curation, Software, Formal analysis, Validation, Investigation, Visualization, Methodology, Writing—original draft, Writing—review and editing; Anthony Randal McIntosh, Resources, Data curation, Formal analysis, Validation, Methodology, Writing—original draft, Writing—review and editing; Viktor Jirsa, Resources, Software, Validation, Methodology, Writing—original draft, Writing—review and editing; Gustavo Deco, Software, Visualization, Methodology, Writing—original draft, Writing—review and editing; Petra Ritter, Conceptualization,

Resources, Data curation, Formal analysis, Supervision, Funding acquisition, Validation, Investigation, Visualization, Methodology, Writing—original draft, Project administration, Writing—review and editing

### Author ORCIDs
Michael Schirner (iD) http://orcid.org/0000-0001-8227-8476
Viktor Jirsa (iD) http://orcid.org/0000-0002-8251-8860
Petra Ritter (iD) http://orcid.org/0000-0002-4643-4782

### Ethics

Human subjects: Research was performed in compliance with the Code of Ethics of the World Medical Association (Declaration of Helsinki). Written informed consent was provided by all subjects with an understanding of the study prior to data collection, and was approved by the local ethics committee in accordance with the institutional guidelines at Charité Hospital Berlin.

### Decision letter and Author response

Decision letter https://doi.org/10.7554/eLife.28927.022
Author response https://doi.org/10.7554/eLife.28927.023

## Additional files

### Supplementary files

• Supplementary file 1. Supplementary table to the eLife Transparent Reporting Form that summarizes the used statistical test, N, exact p-values and descriptive statistics for each hypothesis test.
DOI: https://doi.org/10.7554/eLife.28927.017

• Transparent reporting form
DOI: https://doi.org/10.7554/eLife.28927.018

### Major datasets

The following dataset was generated:

| Author(s) | Year | Dataset title | Dataset URL | Database, license, and accessibility information |
| --- | --- | --- | --- | --- |
| Schirner M, McIntosh AR, Jirsa V, Deco G, Ritter P | 2017 | Hybrid Brain Model data | http://dx.doi.org/10.17605/OSF.IO/MNDT8 | Available at Open Science Framework Repository under a CC0 1.0 Universal license |

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
