## [Decision Letter]

Thank you for submitting your article "Inferring multi-scale neural mechanisms with brain network modelling" for consideration by *eLife*. Your article has been favorably evaluated by David Van Essen (Senior Editor) and three reviewers, one of whom, Charles E. Schroeder (Reviewer #1), is a member of our Board of Reviewing Editors. The following individual involved in review of your submission has agreed to reveal their identity: Karl J Friston (Reviewer #2).

The reviewers have discussed the reviews with one another and the Reviewing Editor has drafted this decision to help you prepare a revised submission.

Summary:

The authors introduce a novel connectome-based brain network model that in their view integrates structural and functional data from individuals with neural population dynamics to enable what they term "multi-scale neurophysiological inference." The most novel feature is that in-silico populations are linked together by empirically derived structural connectivity and driven by electroencephalography (EEG) source activity (i.e., the source activity functions. The authors report that simulations predicted subjects' individual resting-state functional magnetic resonance imaging (fMRI) time series and spatial network topologies over 20 minutes of activity, but more importantly, they also revealed precise neurophysiological mechanisms that underlie and link observations from multiple scales and measurement domains including a) r-fMRI oscillations, b) topographies of macroscopic FC networks, c) E/I balance at a more cellular level, d) α-related "pulsed inhibition" and e) power-law scaling in fMRI. The authors offer the model as a potential solution to bridging the gap between temporal/spatial scales and measurement domains. The strengths of the paper include the fact that the simulations predicted the fMRI time series better than other control scenarios. The use of EEG derived temporal functions to drive the models seems a good step in the direction of biological plausibility. Interestingly, the authors observed several neurophysiological phenomena that were produced by the model. The analysis seems to be solid and rigorous. However, since there were many details in each analysis, the organization of the manuscript and the description and motivation of each step could be improved.

Essential revisions:

1) The idea to inject empirical data into a biophysical model is very appealing, although there is a logical conundrum. The model itself (without IBG, the EEG currents) is already internally consistent – i.e. even without the injected EEG data, it can predict both EEG (through EPSP-related currents Ii(E), Equation (1)) and bold, through convolution of the same with the HRF. But here, the empirical data is already used as part of the input to the pyramidal cells. That is, the currents Ii(E)of Equation (1) are composed, by construction of the self-consistent feedback from other parts of the modelled system plus the actual behaviour of the system IBG. As soon as the system itself exhibits some internally generated behaviour (which it seems to), then the model currents Ii(E)receive inputs from other regions and can no longer predict the EEG which is now just part of the input to the RHS of Equation (1) – i.e. the model loses self-consistency. Part of the circularity arises when comparing the predicted EEG from Ii(E) with the actual EEG IBG, clearly already a subcomponent of the system. Notably, only when the coupling is turned off (subsection “Long-range coupling controls fMRI power-law scaling”, second paragraph) does the PSD of the model match the PSD of the EEG data.

Similarly, the EEG data are themselves already generated by a highly-integrated system (i.e. the brain) and there is thus a similar circularity between its biological generation, and then the apparent additional contributions from the model. That is, whether you start with the model and add EEG, or start with EEG and add model, either is contradictory because they're meant to be the same thing.

The only other alternative, is that the model itself essentially does nothing, in which case, its own contributions to Ii(E) are negligible so that Ii(E)IBG. This could arise if the inputs drive essentially strong inhibitory activity, damping the behaviour of the system such that all of the activity is essentially driven by IBG. This is partly supported by the biggest match with data occurring for increasing wBG(I), Figure 3 – i.e. the drive to the inhibitory population favouring a strong EEG input. Although self-consistency would be approximated in this setting, this would be essentially a trivial undertaking. Regardless, Figure 5–Figure 6 suggest this is not the case – i.e. the model is driven away from damped fixed point.

What is the ratio of the model terms on the RHS of Equation (1) to wBG(E)IBG?

Note: in discussion, this emerged as the largest single concern, and the onus lies on the authors resolve this apparent logical inconsistency.

2) It seems highly unlikely that the "standard model" driven by random fluctuations has any chance of yielding time series that correlate with the actual BOLD correlations – even if it is a really good model of the brain, i.e. with accurate parameters etc., it is driven by completely random inputs – how could these yield accurate time-resolved BOLD fluctuations? If this is the criteria by which a goodness of fit is being evaluated, then an alternative inversion scheme, such as dynamic expectation maximization or generalized particle filtering [1] – which can infer the true hidden stochastic inputs – should have been employed. Otherwise it is no wonder that the correlations are close to zero. To some extent, the same also goes for the permutation model which is given the wrong driving inputs, even if they are presumably correlated with the actual source reconstructed data for each region. When it comes to the static (time averaged) functional connectivity, Figure 4, the permutation and noise models perform reasonably well – this being pertinent because the noise model has far fewer parameters and the comparison, based on linear correlations and not model likelihood, carries no complexity penalty (as it ideally should).

3) Although the fitting of the PSD is state-of-the art, a linear fit across one order of magnitude is not sufficient evidence of scale-free dynamics (subsection “Long-range coupling controls fMRI power-law scaling”, first paragraph) which typically require broader scaling and may be better studied by measuring the distribution of fluctuations in the time domain. In the PSD, power law requires a particular slope (the noise becomes white when the slope becomes shallow). Criticality really requires analysis for a critical transition when tuning a model parameter. The slope is still a useful measure of model performance but the language around it should be curtailed.

4) In what dynamical regime is the simulated neural mass model? Is it near-to-criticality (as confirmed by analysis of the effective Jacobian) as per some of the authors' prior work. If so, what is the nature of the bifurcation? Shouldn't the injected noise (in the noise model) have the same autocorrelation as the source EEG data to allow like with like? I feel there is a lot of back literature on this model (e.g. by Wang) as well as neural mass dynamics in general that would improve the contextualization of the paper.

5) What is the motivation for choosing the "best" model as the one that best "converts" the EEG input to an fMRI output? On first thought, that makes complete sense but this is not what the brain is trying to do (?).… can we still observe the neurophysiological phenomenon if the selected parameters weren't the ones that showed the highest correlation with the fMRI? Perhaps using the model with a cognitive task might be informative and also provide some details about behavioral performance.

6) Level at which the paper is presented: On one hand, there seems to be an attempt to present this at a level appropriate for the broad readership of *eLife*. On the other hand, it seems that in many cases the descriptions lack precision and technical terms seem to be used freely with no attempt to introduce them. There are lots of repetitions in the text e.g., "Individualized hybrid models yielded predictions of ongoing empirical subject-specific resting-state fMRI time series." Figures are graphically nice but captions are messy; also, there is a loose relation between what they refer to in the text and what you see in figures). Introduction is more of an extended Abstract.

7) The hybrid model as presently constructed is specifically tied to α oscillatory dynamics. This could be a limitation, as α, though prominent, is different in many ways from other oscillatory regimes, and α is certainly not dominant at all times. In fact, it may be best thought of as one of several modes of brain operation, even in a system that is for lack of a better description, at "rest."

8) Regarding clarifying the presentation – The three issues you should emphasise could be summarised by inserting something like the following in the Introduction: "In summary, our biophysically grounded (whole brain) model has the potential to test mechanistic hypotheses about emergent phenomena such as scale-free dynamics, the crucial role of excitation-inhibition balance, the haemodynamic correlates of α activity etc. However, there is another perspective on this form of hybrid modelling. Because it uses empirical EEG data to generate predictions of fMRI responses, it can be regarded as a form of multimodal fusion; under a forward or generative model that is both physiologically and anatomically grounded. In addition, because we use connectivity constraints based on tractography, it also serves to fuse structural with functional data."

In the Discussion something like this might help: "Clearly, the sequence of analyses and implicit hypothesis testing presented in this paper represents one of many lines of enquiry. The more general point made by this report is that our hybrid model can be used to both test hypotheses and to build hypotheses. In other words, many of the questions (for which we offer answers) only emerged during application of the model – an application that allowed us to pursue a particular narrative in understanding the genesis of scale free dynamics in the human brain."

The third point should be addressed after the discussion about scale invariant dynamics (Discussion, seventh paragraph). I would recommend something like: "Note that we have used the hybrid model not simply to establish the prevalence of scale invariant dynamics – but to use the power law scaling in a quantitative sense to understand the mechanisms leading to particular power law exponents; for example, the importance of extrinsic (between node) connections in explaining the differences between power law scaling at the electrophysiological and haemodynamic level. This is an important point because scale-free behaviour per se would be difficult to avoid in simulations of this sort."

9) The captions are unwieldly and difficult to track. they should be shortened and edited to describe what is in the figure at hand. Some of the captions, particularly the first few, are quite lengthy, with references, editorializing and even discussion.

[Editors' note: further revisions were requested prior to acceptance, as described below.]

Thank you for resubmitting your work entitled "Inferring multi-scale neural mechanisms with brain network modelling" for further consideration at *eLife*. Your revised article has been favorably evaluated by David Van Essen (Senior Editor), a Reviewing Editor, and a reviewer.

The manuscript has been improved but there are five remaining issues that need to be addressed before acceptance, as outlined below:

1) Self-consistency/causal circularity: yes I agree that injecting EEG as an input current is a great idea (as I did before) unless one is actually predicting that EEG. Finding a less than perfect correlation because the predicted EEG also have local currents on top of the actual injected EEG really just underlines the conceptual issue.

2) The rather unlikely comparisons of the actual (time resolved) fMRI using the actual (time resolved) EEG as an input compared to using time permuted EEG or white noise is still an odd thing to do, whether it's a formal model comparison or an "informal" one. Comparing the moments of the data would be fine. I guess it just shows that "something is working".

3) Since the authors have qualified that their model is focused on the α rhythm, it would be nice if they qualified how it contrasts with the bimodal/bistable corticothalamic neural field model of Freyer et al. (2009,2011) since this has been previously shown to substantially constrain model space (and is a clear feature of the α rhythm not addressed here, but could be done in future work.

Note – Two additional points of concern were triggered by the following passage and resulting revisions of the manuscript text: Both, the approximation of EPSCs on the basis of EEG, and the prediction of EEG on the basis of EPSCs, are in accordance with empirical and theoretical results that identify EPSCs as the major generators of EEG (Buzsáki et al., 2012; Kirschstein and Köhling, 2009; Mitzdorf, 1985). Although EEG reflects the total sum of all extracellular currents, "large cortical pyramidal neurons in deep cortical layers play a major role in the generation of the EEG" and, more importantly, "excitatory postsynaptic potentials predominate as generators of the EEG waves", since "CI- and K^+^ as main charge carriers of the IPSP have a smaller electrochemical gradient than Na^+^ and Ca^2+^ (the charge carriers of the EPSP)" (Kirschstein and Köhling, 2009), while IPSPs do not contribute significantly to current source densities(Mitzdorf, 1985).

4) First – regarding the proposition (Mitzdorf, 1985) that you can basically ignore the IPSP as a contributor to local CSD profile (and to the EEG): theoretical considerations notwithstanding, the literature on CSD in nonhuman has repeatedly presented empirical observations that are exceptions to this proposition [for an early review addressing ERP generators, see (Schroeder et al., 1995)]. Also the CSD correlates of entrainment (Lakatos et al., 2008; Lakatos et al., 2013) clearly reflect temporal alternation of states dominated by net ensemble depolarization and hyperpolarization (part of which reflects IPSCs), both of which clearly impact the CSD profile and are associated fluctuations in neuronal firing.

5) Second – regarding the role of the large deep (Layer 5) pyramids in the EEG: it does seem that particular large Layer 5 cells like intrinsic burst neurons are critical to the orchestration of oscillations [e.g., (Carracedo et al., 2013; Vijayan et al., 2015; Sherman et al., 2016)]. However, the papers that have examined the laminar CSD profiles associated with δ, theta and α oscillations (Lakatos et al., 2005; Bollimunta et al., 2008; Bollimunta et al., 2011; Haegens et al., 2015) generally show that while all the layers are involved in generating EEG oscillations, the largest generator currents are in the supragranular layers. The layer 5 pyramids to have apical dendrites there, but the majority of the synapses in the supra layers are on supragranular pyramids.

On the other hand, the reviewers agree is a tremendous amount of work in this paper and, that despite these concerns, the balance is strongly positive.

---

## [Author Response]

Essential revisions:1) The idea to inject empirical data into a biophysical model is very appealing, although there is a logical conundrum. The model itself (without IBG, the EEG currents) is already internally consistent – i.e. even without the injected EEG data, it can predict both EEG (through EPSP-related currents Ii(E), Equation (1)) and bold, through convolution of the same with the HRF. But here, the empirical data is already used as part of the input to the pyramidal cells. That is, the currents Ii(E) of Equation (1) are composed, by construction of the self-consistent feedback from other parts of the modelled system plus the actual behaviour of the system IBG. As soon as the system itself exhibits some internally generated behaviour (which it seems to), then the model currents Ii(E)receive inputs from other regions and can no longer predict the EEG which is now just part of the input to the RHS of Equation (1) – i.e. the model loses self-consistency. Part of the circularity arises when comparing the predicted EEG from Ii(E)with the actual EEG IBG, clearly already a subcomponent of the system. Notably, only when the coupling is turned off (subsection “Long-range coupling controls fMRI power-law scaling”, second paragraph) does the PSD of the model match the PSD of the EEG data.Similarly, the EEG data are themselves already generated by a highly-integrated system (i.e. the brain) and there is thus a similar circularity between its biological generation, and then the apparent additional contributions from the model. That is, whether you start with the model and add EEG, or start with EEG and add model, either is contradictory because they're meant to be the same thing.The only other alternative, is that the model itself essentially does nothing, in which case, its own contributions to Ii(E) are negligible so that Ii(E)~IBG. This could arise if the inputs drive essentially strong inhibitory activity, damping the behaviour of the system such that all of the activity is essentially driven by I_BG_. This is partly supported by the biggest match with data occurring for increasing wBG(I), Figure 3 – i.e. the drive to the inhibitory population favouring a strong EEG input. Although self-consistency would be approximated in this setting, this would be essentially a trivial undertaking. Regardless, Figure 5–Figure 6 suggest this is not the case – i.e. the model is driven away from damped fixed point.What is the ratio of the model terms on the RHS of Equation (1) to wBG(E)IBG?Note: in discussion, this emerged as the largest single concern, and the onus lies on the authors resolve this apparent logical inconsistency.

We thank the reviewers for drawing our attention to this important issue. The hybrid model is clearly limited, because it is not an autonomous (self-contained) model of the brain, but is dependent on externally injected activity. However, we would like to argue that input-injection does not interfere with the validity of the simulated population outputs or the ability of the model to make meaningful predictions. Similar to in vitro or in vivo electrophysiology current injection experiments, the response of the neural model (be it physical or virtual) to stimulation allows performing controlled experiments, building and testing hypotheses. The validity of model outputs remains, because the logic and validity of population inputs remains. That is, the hybrid model retains the self-consistency of the inputs of the original model, because injected EEG is not introduced as an additional term to the sums of input currents Ii(E) and Ii(I), but as a replacement for the terms for local excitatory connections. That is, we removed the term w_+ * J_NMDA * S_i^(E) from the RHS of equation (5) and the term J_NMDA * S_i^(E)from the RHS of Equation (6) from the original model in Deco et al. (2014) and replaced them with the terms wBG(E)IBG (Equation 1, our manuscript), respectively wBG(I)IBG (Equation 2). Injected EEG is used as a biologically plausible approximation of local excitatory synaptic currents (EPSCs) mediated by local connections. Therefore, EEG injection does not disrupt the self-consistency of the original equations, because the original configuration of input currents (i.e. local excitatory, local inhibitory, and long-range excitatory) is maintained and EEG is used as an approximation of only one subcomponent of input currents (EPSCs). Furthermore, self-consistency with regard to the prediction of EEG is also retained: EEG-predicting forward models that use not the entire sum of input currents, but only EPSCs, would again generate the EEG that was previously injected. Both, the approximation of EPSCs on the basis of EEG, and the prediction of EEG on the basis of EPSCs, are in accordance with empirical and theoretical results that identify EPSCs as the major generators of EEG (Buzsáki et al., 2012; Kirschstein and Köhling, 2009; Mitzdorf, 1985). Although EEG reflects the total sum of all extracellular currents, “large cortical pyramidal neurons in deep cortical layers play a major role in the generation of the EEG” and, more importantly, “excitatory postsynaptic potentials predominate as generators of the EEG waves”, since “CI^-^ and K^+^ as main charge carriers of the IPSP have a smaller electrochemical gradient than Na^+^ and Ca^2+^ (the charge carriers of the EPSP)” (Kirschstein & Köhling, 2009), while IPSPs do not contribute significantly to current source densities (Mitzdorf, 1985). This approach is however clearly limited by the coarse spatial resolution of EEG and our inability to differentiate between the local and long-range components of EPSCs. That is, we cannot disentangle the individual contributions from local and global connectivity. As a result, the sum of both excitatory current terms (local and global) is not identical anymore to the injected EEG source activity (the average correlation between the sum of excitatory currents and injected EEG is however still r=0.58) and hence after forward modeling a discrepancy between simulated and injected EEG would occur. This is, however, not a conceptual issue but rather a limitation that arises from the aforementioned superposition of local and global currents in the EEG; the limitation wouldn’t exist if it would be possible to perfectly disentangle local from global EPSCs. We therefore emphasize that EEG is merely an *approximation* of the true synaptic input currents, which, however, is able to generate meaningful predictions.

Referring to the comment “Notably, only when the coupling is turned off (subsection “Long-range coupling controls fMRI power-law scaling”, second paragraph) does the PSD of the model match the PSD of the EEG data”, we would like to note that the comparison of PSD power-law exponents involves two different signals: the exponent of simulated fMRI amplitude fluctuation (not from EEG) is compared to the exponent of empirical EEG band-power fluctuations. In this analysis we ask why the exponent of empirical fMRI (β = -0.83, He et al. 2011, respectively β = -0.82 in our analysis) is larger than that of empirical EEG (β = -0.47). We asked this question because our results showed a mechanism that transformed α-band power oscillations into fMRI amplitude oscillations. In light of the identified mechanism it was surprising that the exponent of α-band fluctuations was smaller than that of the generated fMRI amplitude fluctuations. Interestingly, however, when large-scale coupling was deactivated, we found that the exponent of simulated fMRI matched the exponent of EEG. With everything else being identical, we concluded that long-range coupling was responsible for increasing the exponent of simulated fMRI. We tried to emphasize this point better by improving the wording in the respective Results section.

The following table shows the ratios of the model terms on the RHS of Equation (1) to wBG(E)IBG (median ratios of standard deviations over all regions and subjects; median was used because ratio distributions were heavily skewed due to regions where injected activity had extremely low s.d.s that led to high ratio values). The table indicates that all three time-varying terms have non-negligible contributions to the total sum of input currents.

s.d. ratio (column divided by row)WEIOG∑jCijSj(E)JiSi(I)wBG(E)IBG0 (constant)1.054.55

Table 1. Ratios of model terms.

The importance of long-range coupling can also be seen by comparing Figure 6 with Figure 7: long-range input (in the otherwise identical model) has a considerable impact on the amplitudes of firing rates, synaptic gating and fMRI on longer time scales.

To address the aforementioned points, we added the following paragraph to the Discussion section.

“The idea of the hybrid approach is to test how biophysically based and structurally constrained models respond to biologically plausible synaptic input currents, comparable to in vivo or in vitro electrophysiology current injection experiments. […] This is, however, not a conceptual issue but rather a limitation that arises from the aforementioned superposition of local and global currents in the EEG.”

2) It seems highly unlikely that the "standard model" driven by random fluctuations has any chance of yielding time series that correlate with the actual BOLD correlations – even if it is a really good model of the brain, i.e. with accurate parameters etc., it is driven by completely random inputs – how could these yield accurate time-resolved BOLD fluctuations? If this is the criteria by which a goodness of fit is being evaluated, then an alternative inversion scheme, such as dynamic expectation maximization or generalized particle filtering [1] – which can infer the true hidden stochastic inputs – should have been employed. Otherwise it is no wonder that the correlations are close to zero. To some extent, the same also goes for the permutation model which is given the wrong driving inputs, even if they are presumably correlated with the actual source reconstructed data for each region. When it comes to the static (time averaged) functional connectivity, Figure 4, the permutation and noise models perform reasonably well – this being pertinent because the noise model has far fewer parameters and the comparison, based on linear correlations and not model likelihood, carries no complexity penalty (as it ideally should).

We share the view of the reviewers that it seems highly unlikely that the original brain model yields time series that correlate with actual empirical BOLD time series. Indeed, this is the very point that we are trying to make when comparing hybrid modeling results with the three control scenarios: we wanted to show that the hybrid model is able to predict fMRI time series while the original model and the random permutation model are not. Our intention here was to estimate whether the standard/permutation models are able to predict fMRI *at all*; the intention was not to make claims in the sense of formal model comparison. Although it may seem unlikely that a noise-driven model is able to predict actual empirical fMRI, we nevertheless wanted to proof that point to the reader. The comparison with the original models serves as a sort of “baseline”, which shows that the achieved prediction quality of the hybrid model is actually noteworthy and that the original brain model, in the same configuration as it was used in previous studies, does not produce these correlations. Furthermore, we wanted to test whether it is the specific temporal sequence of time points present in the injected activity that made this prediction possible. To make this point, we performed the test with permuted input time series.

To better acknowledge the point raised in this comment, we added the following paragraph to the revised manuscript:

“We note that our comparison of prediction qualities of the hybrid model and the three control scenarios is not a result in the sense of formal model comparison where goodness of fit is assessed in light of model complexity. […] To make this point, we performed the test with permuted input time series.”

Although in the case of the noise model one parameter less (i.e. two) was tuned as in the case of the hybrid model (three), we do not penalize complexity. While minimizing complexity is one pillar of formal model comparison, we focus here on maximizing the biological plausibility of the model. For example, the α regressor is a very simple model that contains zero free parameters, but nevertheless yields comparable fMRI time series correlations as the hybrid model. In the case of identical (or similar) fMRI time series prediction accuracies, an estimator of relative model quality like AIC would favor the simpler model (i.e. the α regressor) over the more complex model (i.e. the hybrid model), despite the fact that the α regressor does not account for the different sorts of neural phenomena for which the hybrid model accounts for. The α regressor clearly lacks the ability to account for the range of different empirical phenomena (Figure 2) that the hybrid model accounts for, as it is not based on a set of different state variables that correspond to biological entities as the hybrid model is. To formally compare the quality of noise model, α regressor and hybrid model, it would not be sufficient to base model likelihood on the accuracy of isolated goodness-of-fit metrics of individual state variables, e.g. “fMRI time series similarity” or “excitation-inhibition ratio of input currents” or “phase relationship of LFP α cycles vs. firing rates”. We are currently working towards incorporating formal model comparison into the hybrid modeling framework to be able to compute likelihood functions that estimate a model’s quality in concurrently predicting the dynamics of empirical phenomena in different modalities, metrics and at different levels (e.g. the quality of simultaneous predictions of fMRI time series, oscillatory electric activity, phase relationships between LFP and firing, and excitation-inhibition balance ratios of input currents).

3) Although the fitting of the PSD is state-of-the art, a linear fit across one order of magnitude is not sufficient evidence of scale-free dynamics (subsection “Long-range coupling controls fMRI power-law scaling”, first paragraph) which typically require broader scaling and may be better studied by measuring the distribution of fluctuations in the time domain. In the PSD, power law requires a particular slope (the noise becomes white when the slope becomes shallow). Criticality really requires analysis for a critical transition when tuning a model parameter. The slope is still a useful measure of model performance but the language around it should be curtailed.

Scale-invariance was not, as indicated by the comment, determined by PSD fitting. Only the power-law exponent was computed in the PSD (using a power-law function) to ensure comparability of our estimates with those from previous empirical work (He, 2011).

“Finally, to obtain the power-law exponent β, the <0.1 Hz range of each average power spectrum was fit with a power-law function: P(f) ∝ 1/f^β^ using a least-squares fit. Using the low-frequency range to fit the power-law exponent avoids aliasing artifact in higher-frequency range (we used TR of 2.16 s, hence Nyquist limit is 0.23 Hz) and yields reliable measurement of the scale-free distribution (Eke et al., 2002).” (from He (2011))

As the comment recommends, we determined scale-freeness by measuring the distribution of fluctuations in the time domain using a variant of dynamic fluctuation analysis in order to determine scale-freeness:

“Time series were tested for scale invariance using rigorous model selection criteria that overcome the limitations of simple straight-line fits to power spectra for estimating scale invariance (see Materials and methods; for illustration purposes straight-line fits are shown in Figure 7 and Figure 7—figure supplement 1).”

“To test for the existence of scale invariance we used a method that combines a modified version of the well-established detrended fluctuation analysis (DFA) with Bayesian model comparison (Ton & Daffertshofer, 2016). DFA is, in contrast to PSD analyses, robust to both stationary and nonstationary data in the presence of confounding (weakly non-linear) trends.”

(lines 965 to 969 in the original manuscript)

We clarified this point further in the revised manuscript:

“straight-line fits of power spectra are for illustration purposes only; scale-invariance was determined in the time domain using rigorous model selection criteria, see Materials and methods”.

“We tested for the existence of power-law scaling in the time domain by using rigorous model selection criteria that overcome the limitations of simple straight-line fits to power spectra (see Materials and methods; for illustration purposes straight-line fits are shown in Figure 7 and Figure 7—figure supplement 1).”

We further clarified in the Discussion that power-laws alone are not sufficient to establish criticality:

“Central to theories on the emergence of criticality is the tuning of a control parameter (e.g. connection strengths) that leads the system to a sharp change in one or more order parameters (e.g. firing rates) when the control parameter is moved over a critical point that marks the boundary of a phase transition. […] Rather, criticality requires the existence of a control parameter that can be adjusted to move the system through a phase transition at a critical point (Beggs & Timme, 2012).”

4) In what dynamical regime is the simulated neural mass model? Is it near-to-criticality (as confirmed by analysis of the effective Jacobian) as per some of the authors' prior work. If so, what is the nature of the bifurcation? Shouldn't the injected noise (in the noise model) have the same autocorrelation as the source EEG data to allow like with like? I feel there is a lot of back literature on this model (e.g. by Wang) as well as neural mass dynamics in general that would improve the contextualization of the paper.

The parameter values of the original (noise-driven) neural mass model were taken from Deco et al. (2014). There, bifurcation diagrams (Figure 2) for the parameter G indicate that below a critical value of G, the optimization of feedback inhibition weights (FIC) makes the network converge to a single stable state of low firing activity. For larger values of G, long-range interactions are too strong to be compensated by FIC and the activity diverges and jumps to a high activity state. In our present study the noise-driven model played a minor role and was only used to show that the achieved correlation between simulated and empirical fMRI time series is not a trivial outcome. For performing the comparison between hybrid and noise model we tuned the G and J_NMDA parameters of the noise model well through the region where low activity switches to high activity.

Since our hybrid model has a time-dependent term it is a non-autonomous system of the form dxdt=fx+υ(t) or equivalently ddtxτ=fx+υ(τ)1.

The “state space” of such a system does not have any equilibria becauseτ˙=1≠0, i.e. there are no bifurcations, since the number or stability of equilibria does not change.

The comment on the autocorrelation of injected noise seems to target a similar point as comment #2. Therefore, we would like to reiterate that our intention for showing this result was to enable comparison between hybrid model results and the original model from which it was derived and to show that the prediction of fMRI time series is a novel aspect of the hybrid model and not something that was already existent in the original noise-driven model.

5) What is the motivation for choosing the "best" model as the one that best "converts" the EEG input to an fMRI output? On first thought, that makes complete sense but this is not what the brain is trying to do (?).… can we still observe the neurophysiological phenomenon if the selected parameters weren't the ones that showed the highest correlation with the fMRI? Perhaps using the model with a cognitive task might be informative and also provide some details about behavioral performance.

Our motivation for choosing that parameter set that produces the highest correlation between simulated and empirical fMRI time series is based on our goal to infer the underlying (but unobservable) behavior and parameters of the real system. This idea is based on the assumption that when the model optimally fits observable brain activity, then also the underlying unobservable brain activity is faithfully reproduced. Our goal is to have a model that better explains the specific dynamics of ongoing subject-specific BOLD activity. Previous brain modeling approaches mainly focused on stationary features of brain activity, like functional connectivity or frequency composition; however, reproduction of FC does not necessarily imply that also the underlying time series show biologically plausible behavior. The motivation for the model was not to have a simple “converter”. For mere “conversion” of EEG into fMRI, a simple and abstract model (like the α regressor) might be better suited. Rather, the intention was to use EEG as an approximation of ongoing subject-specific synaptic currents in order to infer the (non-invasively unobservable) physiological processes that underlie signals like EEG and BOLD. For a given model we consider brain activity to be optimally explained when the fit between predicted and empirical activity is highest. The model is not a phenomenological model of brain function (i.e. “what the brain is *trying* to do”) but aims to understand the physiological processes that are implemented in neural circuits and that underlie the observed emerging activity (i.e. “how are the different elements of a brain acting together to produce emerging behavior?”). The state variables and parameters of this model relate to biophysical variables and parameters. Therefore, our motivation for maximizing the fit between empirical and simulated data was to optimally approximate the true underlying (but unobservable) behavior and parameters of the real system.

We further emphasized this point in the revised manuscript:

“Our motivation for choosing that parameter set that produces the highest correlation between simulated and empirical fMRI time series is based on our goal to infer the underlying (but unobservable) behavior and parameters of the real system. This idea is based on the assumption that when the model optimally fits observable brain activity, then also the underlying unobservable brain activity is faithfully reproduced.”

And

“This approach is motivated by the need for a model that not only reproduces static features of brain activity, like functional connectivity, but that produces these features on the basis of biologically plausible time series dynamics. […] Here, imperfect reproduction of neural activity directly points to deficits in our understanding and conceptualization of large-scale brain structure and function, which to iteratively improve is the goal of this approach.”

To test whether the neurophysiological phenomena also emerge for other parameter sets we sampled the parameter space at six different positions and tested for emergence of the phenomena (testing and evaluating a high number of parameters is computationally demanding and also beyond the scope of our study). We found that either not all mechanisms were simultaneously reproduced or that they were less pronounced. The following tables summarize our results.

(E/I balance was determined by correlation between EPSC and IPSC)

GwBG(E)wBG(I)wBG(E)Best set0.120.155Test set 10.120.055Test set 20.120.15150Test set 30.40.155Test set 40.40.055Test set 50.40.05150Test set 60.120.150.5

Table 2. Tested parameter sets.

Firing vs. α cycle [deg]E/I balance [r]Firing vs. α power [r]Α power vs. fMRI [r]Power-law exponent [β]Functional connectivity [r]Best set138°-0.7-0.52-0.45-0.730.47Test set 1156°-0.57-0.32-0.27-1.070.43Test set 2156°-0.25-0.16-0.14-0.140.34Test set 3149°-0.47-0.03-0.02-1.020.35Test set 4160°-0.27-0.07-0.05-1.420.3Test set 5160°-0.050.260.22-1.160.31Test set 626°-0.850.580.54-0.430.11

Table 3. Reproduction of empirical phenomena.

We agree that using the model with task data might be informative and this is the subject of an upcoming study with this model, but is out of the scope of this paper.

6) Level at which the paper is presented: On one hand, there seems to be an attempt to present this at a level appropriate for the broad readership of eLife. On the other hand, it seems that in many cases the descriptions lack precision and technical terms seem to be used freely with no attempt to introduce them. There are lots of repetitions in the text e.g., "Individualized hybrid models yielded predictions of ongoing empirical subject-specific resting-state fMRI time series." Figures are graphically nice but captions are messy; also, there is a loose relation between what they refer to in the text and what you see in figures). Introduction is more of an extended Abstract.

We refined our descriptions and now introduce technical terms (e.g., “Brain network model”, “structural connectivity/connectome”, “source activity”, “functional connectivity”). We removed repetitions that weren’t essential for clarity of the presentation. As per other comments, we have shortened captions to increase the focus on the figure content and we have edited figure captions to improve their readability; references, editorializing and discussion were removed.

7) The hybrid model as presently constructed is specifically tied to α oscillatory dynamics. This could be a limitation, as α, though prominent, is different in many ways from other oscillatory regimes, and α is certainly not dominant at all times. In fact, it may be best thought of as one of several modes of brain operation, even in a system that is for lack of a better description, at "rest."

We would like to address this point noting that the hybrid model itself is not tied to the α rhythm, as the injected EEG source activity was not limited to the α band, but that it was merely our analysis of the initial application of that approach where α rhythms played an important role (in addition to the topics “excitation-inhibition balance” and “scale-freeness”). The hybrid modeling approach does not set any requirements on the frequency spectrum of the injected source activity, it can in principle encompass the entire frequency spectrum of EEG (or MEG) recordings. In this manuscript injected activity was band-pass filtered between 1 and 30 Hz (extending beyond the α band) in order to better exclude artifacts that result from the acquisition of EEG simultaneously with fMRI. This was an additional step in addition to performing standard artifact correction procedures for simultaneous EEG-fMRI data as outlined in the Methods section. It is also worth noting that our focus on α rhythms was not “by design” or “by construction”, but emerged from the sequence of analyses that we performed on the results of our initial application of the hybrid model as we outlined in the Introduction:

“Upon finding that the hybrid model predicts fMRI activity, we first sought to identify how injected EEG drove the prediction of subject-specific fMRI time series. […] Analysis led us to the identification of an inhibitory effect resulting from the interaction of postsynaptic current oscillations and local population circuitry.”

Courtesy of another comment, we now write in the Discussion section that this sequence clearly represents only one of many possible lines of enquiry:

“Clearly, the sequence of analyses and implicit hypothesis testing presented in this paper represents one of many possible lines of enquiry. […] In other words, many of the questions (for which we offer answers) only emerged during application of the model, which allowed us to pursue a particular narrative in understanding the genesis of different empirical phenomena.”

Finally, to address this and another comment, we now explicitly address the limitations of focusing on a single frequency band and our reasons for choosing to do so, despite these limitations:

“It is important to note that the observed processes may not be specific to α-oscillations, but may apply also to other frequencies or non-oscillatory signal components, e.g., phase-locked discharge of neurons occurs over a range of frequency bands and is not limited to the α-rhythm (Buzsaki, 2006). […] In this regard it is interesting to note that the time series correlations obtained by the α-regressor and the hybrid model are comparable, which indicates that the α-rhythm was the main driver for the hybrid model’s fMRI time series prediction.”

8) Regarding clarifying the presentation – The three issues you should emphasise could be summarised by inserting something like the following in the Introduction: "In summary, our biophysically grounded (whole brain) model has the potential to test mechanistic hypotheses about emergent phenomena such as scale-free dynamics, the crucial role of excitation-inhibition balance, the haemodynamic correlates of α activity etc. However, there is another perspective on this form of hybrid modelling. Because it uses empirical EEG data to generate predictions of fMRI responses, it can be regarded as a form of multimodal fusion; under a forward or generative model that is both physiologically and anatomically grounded. In addition, because we use connectivity constraints based on tractography, it also serves to fuse structural with functional data."In the Discussion something like this might help: "Clearly, the sequence of analyses and implicit hypothesis testing presented in this paper represents one of many lines of enquiry. The more general point made by this report is that our hybrid model can be used to both test hypotheses and to build hypotheses. In other words, many of the questions (for which we offer answers) only emerged during application of the model – an application that allowed us to pursue a particular narrative in understanding the genesis of scale free dynamics in the human brain."The third point should be addressed after the discussion about scale invariant dynamics (Discussion, seventh paragraph). I would recommend something like: "Note that we have used the hybrid model not simply to establish the prevalence of scale invariant dynamics – but to use the power law scaling in a quantitative sense to understand the mechanisms leading to particular power law exponents; for example, the importance of extrinsic (between node) connections in explaining the differences between power law scaling at the electrophysiological and haemodynamic level. This is an important point because scale-free behaviour per se would be difficult to avoid in simulations of this sort."

We thank the reviewer for these very helpful suggestions and have incorporated them in our revised manuscript.

9) The captions are unwieldly and difficult to track. they should be shortened and edited to describe what is in the figure at hand. Some of the captions, particularly the first few, are quite lengthy, with references, editorializing and even discussion.

We have rigorously shortened and edited the captions to improve their readability. References, editorializing and discussion were removed.

[Editors' note: further revisions were requested prior to acceptance, as described below.]

The manuscript has been improved but there are five remaining issues that need to be addressed before acceptance, as outlined below:1) Self-consistency/causal circularity: yes I agree that injecting EEG as an input current is a great idea (as I did before) unless one is actually predicting that EEG. Finding a less than perfect correlation because the predicted EEG also have local currents on top of the actual injected EEG really just underlines the conceptual issue.3) Since the authors have qualified that their model is focused on the α rhythm, it would be nice if they qualified how it contrasts with the bimodal/bistable corticothalamic neural field model of Freyer et al. (2009,2011) since this has been previously shown to substantially constrain model space (and is a clear feature of the α rhythm not addressed here, but could be done in future work.Note – Two additional points of concern were triggered by the following passage and resulting revisions of the manuscript text: Both, the approximation of EPSCs on the basis of EEG, and the prediction of EEG on the basis of EPSCs, are in accordance with empirical and theoretical results that identify EPSCs as the major generators of EEG (Buzsáki et al., 2012; Kirschstein and Köhling, 2009; Mitzdorf, 1985). Although EEG reflects the total sum of all extracellular currents, "large cortical pyramidal neurons in deep cortical layers play a major role in the generation of the EEG" and, more importantly, "excitatory postsynaptic potentials predominate as generators of the EEG waves", since "CI- and K^+^ as main charge carriers of the IPSP have a smaller electrochemical gradient than Na^+^ and Ca^2+^ (the charge carriers of the EPSP)" (Kirschstein and Köhling, 2009), while IPSPs do not contribute significantly to current source densities(Mitzdorf, 1985).4) First – regarding the proposition (Mitzdorf, 1985) that you can basically ignore the IPSP as a contributor to local CSD profile (and to the EEG): theoretical considerations notwithstanding, the literature on CSD in nonhuman has repeatedly presented empirical observations that are exceptions to this proposition [for an early review addressing ERP generators, see (Schroeder et al., 1995)]. Also the CSD correlates of entrainment (Lakatos et al., 2008; Lakatos et al., 2013) clearly reflect temporal alternation of states dominated by net ensemble depolarization and hyperpolarization (part of which reflects IPSCs), both of which clearly impact the CSD profile and are associated fluctuations in neuronal firing.5) Second – regarding the role of the large deep (Layer 5) pyramids in the EEG: it does seem that particular large Layer 5 cells like intrinsic burst neurons are critical to the orchestration of oscillations [e.g., (Carracedo et al., 2013; Vijayan et al., 2015; Sherman et al., 2016)]. However, the papers that have examined the laminar CSD profiles associated with δ, theta and α oscillations (Lakatos et al., 2005; Bollimunta et al., 2008; Bollimunta et al., 2011; Haegens et al., 2015) generally show that while all the layers are involved in generating EEG oscillations, the largest generator currents are in the supragranular layers. The layer 5 pyramids to have apical dendrites there, but the majority of the synapses in the supra layers are on supragranular pyramids.

Taking into account comments (1) and (3-5), we have revised the text which now reads as follows:

“The idea of the hybrid approach is to test how biophysically based and structurally constrained models respond to biologically plausible synaptic input currents, comparable to in vivo or in vitro electrophysiology current injection experiments. […] Extending from these results, future BNM studies could systematically investigate the role of autocorrelated compared to Gaussian inputs and their impact on emerging fMRI dynamics (like FC dynamics), especially since inputs like the EEG source activity used in our hybrid model better capture the autocorrelation structure of biological source currents, which are different from white noise (Haider et al., 2016; Okun et al., 2010).”

2) The rather unlikely comparisons of the actual (time resolved) fMRI using the actual (time resolved) EEG as an input compared to using time permuted EEG or white noise is still an odd thing to do, whether it's a formal model comparison or an "informal" one. Comparing the moments of the data would be fine. I guess it just shows that "something is working".

We agree with the reviewers’ comment (2) that the comparisons between time resolved and time permuted inputs, respectively noise inputs, are unlikely comparisons. However, by showing these correlations we would like to emphasize that the obtained time series correlations of the hybrid model are a non-trivial advancement over the predictive qualities of noise-driven brain network models. To make this point we added two sentences to the start of the paragraph that discusses the motivation for source activity injection, which now reads as follows:

"Hybrid models draw on empirically estimated EEG source activity to constrain synaptic input current dynamics. This approach is motivated by the need for a model that not only reproduces static features of brain activity, like functional connectivity, but that produces these features on the basis of biologically plausible time series dynamics. […] For example, a wide range of waveforms can produce FC-like correlation patterns without necessarily having a biological underpinning."

Furthermore, the comparison serves the purpose to show that the α regressor time series correlations, although on average significantly lower, are not negligible compared to the hybrid model correlations, which underlines the role of the α rhythm as the major generator of fMRI oscillations in the model. More importantly, time series correlations serve as a contrast to FC correlations: while the hybrid model predicts both, time series and FC, the α regressor predicts only time series. Conversely, noise and permuted model do not predict time series, but only FC. It would not be possible to make these points based on moments of the data alone. To better emphasize these points, we revised the part of the Discussion that discusses this issue, which now reads:

"Rather, the informal comparison serves to better assess the hybrid model’s prediction quality in relation to the original model and the α-regressor. […] More importantly, these correlations enable us to show that although noise and permuted input do not produce noteworthy time series correlations, like the α-regressor, they nevertheless predict FC, while the hybrid model predicts both, time series and FC."